# Training-Free Pseudo-Fusion Strategies for Composed Image Retrieval via Diffusion and Multimodal Large Language Models

## Abstract

Composed Image Retrieval (CIR) is an emerging paradigm in Content-based Image Retrieval that enables users to formulate compositional queries by combining a reference image with an auxiliary modality, usually text-based. This approach supports fine-grained search where the target image shares structural elements with the user-provided image but is modified according to the provided auxiliary text. Conventional CIR methods rely on multimodal fusion to combine visual and textual features into a joint query embedding, which requires training modules that align composed queries with the targets. In this work, we propose PeFuse (for pseudo-fusion), a training-free framework that leverages pretrained models to bridge modalities via generative conversion. We introduce two novel strategies: uni-directional and bi-directional conversion, both implemented using diffusion models and multimodal large language models, converting CIR into four single-modality retrieval problems. These methods reformulate CIR as either intra-modal or cross-modal single-query retrieval tasks, bypassing the need for dedicated task-specific training. Extensive experiments on standard benchmarks demonstrate that converting CIR into text-to-image retrieval tasks is more effective than alternative conversion strategies, achieving competitive or superior performance compared with state-of-the-art methods, while maintaining high flexibility thanks to replaceable components of the conversion pipeline. These results highlight the effectiveness of the pseudo-fusion paradigm for composed image retrieval. Our code is publicly available at: https://anonymous.4open.science/r/ComposedImageRetrieval-9241.

## 1 Introduction

Traditional Content-Based Image Retrieval systems allow users to submit image-based queries, bridging the so-called *semantic gap* (Smeulders et al., 2000). This constraint hinders their ability to accommodate nuanced search intents that are inherently multimodal. Composed Image Retrieval (CIR) addresses this limitation by enabling users to formulate a query using a reference image coupled with an auxiliary modality to specify desired modifications. This approach facilitates fine-grained retrieval, which is particularly valuable in domains like e-commerce (Baldrati et al., 2022), digital asset management (Net & Gomez, 2025), and creative design (Song et al., 2025).

CIR introduces a number of challenges. An effective retrieval system must not only comprehend the individual modalities but also model their compositional semantics, capturing how the modification alters the meaning of the reference image. A prevalent solution involves multimodal feature fusion, in which visual and textual representations are integrated into a unified embedding prior to retrieval. Although recent advances in deep Transformer-based architectures (Vaswani et al., 2017; Dosovitskiy et al., 2021) have improved cross-modal alignment and compositional reasoning, the majority of existing methods rely heavily on dedicated training on large-scale, annotated CIR datasets. To alleviate these restrictions, researchers either synthesize triplet datasets (Li et al., 2025; Wang et al., 2025; Xing et al., 2025) or rely on existing image-text pairs (Jiang et al., 2024) to train models. This dependency limits their scalability and adaptability to significant domain shifts, for which zero-shot CIR (ZS-CIR) is regarded as an effective solution.

In this work, we investigate *training-free pseudo-fusion* strategies for ZS-CIR that circumvent the need for additional task-specific fusion. We propose to pseudo-fuse the multimodal query through *uni-directional* and *bi-directional conversion* techniques, leveraging recent advancements in Diffusion Models (Ho et al., 2020; Song et al., 2021; Rombach et al., 2022) and Multimodal Large Language Models (MLLMs) (Liu et al., 2023; Grattafiori et al., 2024; Yang et al., 2025). The uni-directional approach reformulates the multimodal-query CIR task into a standard uni-modal query problem by converting the reference image and text modification into a synthesized image or a composed description. The bi-directional approach extends this by additionally converting the candidate images into text, enabling a text-based matching paradigm. These strategies facilitate flexible adaptation of existing, off-the-shelf retrieval systems without requiring architectural modifications, fine-tuning, or any training.

Extensive experiments on standard CIR benchmarks demonstrate that converting CIR tasks to text-to-image tasks via the proposed framework achieves competitive or superior performance compared to the state-of-the-art (SOTA) models. Our findings underscore the significant potential of training-free approaches in compositional retrieval. To the best of our knowledge, this is the first work to systematically explore and benchmark modality conversion strategies utilizing both diffusion models and MLLMs for ZS-CIR. In summary, our contributions are as follows:

- Versatile training-free pseudo-fusion strategies for ZS-CIR that seamlessly convert multimodal queries into a single modality, enabling compatibility with existing retrieval models without task-specific fine-tuning.

- A systematic study and comprehensive benchmarking of both uni-directional and bi-directional modality conversion paradigms for CIR using diffusion models and MLLMs.

- A quantitative analysis to elucidate the relationship between CIR performance and key hyperparameters of both MLLMs and diffusion models, as well as a component-based latency analysis when utilizing MLLMs and diffusion models to reframe CIR tasks to single-modality retrieval tasks.

- Empirical evidence that reformulating CIR to text-to-image retrieval is (for now) more effective than other tasks, underscoring the substantial opportunity to further enhance image generation quality through diffusion models.

## 2 Related Work

Early CIR approaches like TIRG (Vo et al., 2019) relied on joint embedding spaces trained with contrastive objectives (van den Oord et al., 2018; Chen et al., 2020; He et al., 2020), where the fused image–text representation was directly compared against candidate image embeddings. Subsequent transformer-based methods (Jia et al., 2021; Li et al., 2022; 2023), pretrained on large-scale vision–language datasets, achieved stronger cross-modal alignment. Building on this foundation, Combiner (Baldrati et al., 2022) leverages CLIP (Radford et al., 2021) to compute integrated features from reference images and accompanying textual descriptions.

A notable line of work builds upon the idea of representing images as pseudo-word tokens within a text sequence. Inspired by Textual Inversion (Gal et al., 2023), methods such as SEARLE (Baldrati et al., 2023b), Pic2Word (Saito et al., 2023), and LinCIR (Gu et al., 2024) map reference images into token embeddings that can be processed by language models, achieving SOTA performance through joint training. Other approaches like CLIP4CIR (Baldrati et al., 2024) introduce learnable fusion operators to better capture compositional semantics. More recently, the generative capabilities of diffusion models (Ho et al., 2020; Song et al., 2021; Rombach et al., 2022) have also been explored for CIR. For example, CIG (Wang et al., 2025) uses a pretrained textual inversion network to convert a reference image into tokens that can be employed for target image generation and fused with the query.

Despite these advances, the use of LLMs/MLLMs remains more prevalent. For example, DQU-CIR (Wen et al., 2024) fuses unified textual and visual information extracted via LLMs or VLMs. Notably, Hy-CIR (Jiang et al., 2024) enhances model training by incorporating contrastive learning on synthetic triplets,

demonstrating the efficacy of synthetic supervision. Further advancing this approach, Feng et al. (2024) scales both negative and positive samples for contrastive learning using an MLLM. Furthermore, MRA-CIR (Tu et al., 2025) circumvents error-prone intermediate text generation by using a Multimodal Reasoning Agent to directly construct high-quality triplets from unlabeled images. Similarly, CoLLM (Huynh et al., 2025) mitigates data scarcity by synthesizing training triplets from image-caption pairs using LLMs, enabling multimodal fusion.

A common characteristic of the aforementioned methods is their reliance on ad-hoc training, either on synthetic triplets or existing image-text datasets. Although learnable fusion methods achieve strong performance, their requirement for task-specific training limits flexibility and generalizability to new domains or modalities. To generalize to new datasets, training-free CIR methods have been proposed. CIReVL (Karthik et al., 2024) uses an LLM to refine VLM-generated captions by incorporating the text modification to perform text-to-image retrieval. Similarly, WeiMoCIR (Wu et al., 2025) employs an MLLM to caption candidate target images. Moreover, LDRE (Yang et al., 2024) leverages LLMs to generate multiple captions that capture diverse semantic aspects of reference images conditioned on modifications, and subsequently assigns weights to these generated captions to enhance image retrieval performance. Another relevant framework, ImageScope (Luo et al., 2025), unifies various language-guided image retrieval tasks into a text-to-image retrieval task using descriptions generated by an MLLM. However, it relies on multiple models in stages, leading to cumulative compute and increased inference time. In contrast, we use only a single MLLM or diffusion model instead of multiple LLMs or MLLMs, resulting in a simpler yet effective pipeline for reformulating CIR to other tasks.

Unlike these prior works, which often involve multi-stage text generation or ensembles of multiple models, we reformulate CIR into four high-level tasks and systematically benchmark them using as few as one model, including conversion to one-query intra-modal or cross-modal retrieval tasks, as either text-based or image-based retrieval. By leveraging pretrained diffusion models and MLLMs, our approach offers a flexible, modular, and plug-and-play solution for CIR.

## 3 Methodology

As depicted in Figure 1, our method employs a dual-strategy pipeline. Our objective is to transform multimodal queries, consisting of a reference image and a textual modification, into a unified uni-modal representation, either as a diffusion-generated image or an MLLM-generated textual description. This reformulation enables retrieval models to match the query against either target images or their corresponding MLLM-generated textual descriptions. The *uni-directional conversion* (pink dashed box) facilitates retrieval by projecting the query into a target modality, either by using an MLLM to generate descriptive text from images and modifications or by using a diffusion model to generate images from reference images and MLLM-generated text. Specifically, we use MLLMs to convert reference images plus corresponding modifications to text or diffusion models to generate alternative target images from such composed queries. The *bi-directional conversion* (green dashed box) extends this by subsequently using the MLLM to also project target images into the textual modality, enabling a text-based retrieval process. Namely, we additionally generate text based on target images, and then match with generated text or images from uni-directional conversion.

Formally, let $\mathcal{I}$ and $\mathcal{T}$ be the image and text spaces, respectively. Given a reference image $I_{ref} \in \mathcal{I}$ and a text modification $T \in \mathcal{T}$, the goal is to retrieve target images from the same image space $\mathcal{I}$ that are visually similar to $I_{ref}$ while satisfying the semantic requirements specified by $T$, according to a predefined similarity metric. Specifically, assume we have a retrieval model $\Psi(\cdot)$ that takes both image $I \in \mathcal{I}$ and text $T \in \mathcal{T}$ as inputs, and outputs a similarity score $s = sim(\Psi(I), \Psi(T)) \in \mathbb{R}$ based on extracted embeddings $\Psi(I) \in \mathbb{R}^m$ and $\Psi(T) \in \mathbb{R}^m$, an MLLM $f(\cdot)$ which can generate textual descriptions $T_f$ based on an arbitrary combination of images $I$ and text $T$ given the proper dataset-specific prompt $p_d$, and a diffusion model $g(\cdot)$ that generates image $I_g$ based on both images $I$ and text $T$. For CIR, a reference image $I_{ref}^i \in \mathcal{I}$ with corresponding text modification $T_i \in \mathcal{T}$ are paired as a query $(I_{ref}^i, T^i)$, to find the most relevant candidate images in dataset $\mathcal{D} = \{I_{tar}^1, I_{tar}^2, \ldots, I_{tar}^n\}$ based on (cosine) similarity scores:

$$\cos(\Psi(I), \Psi(T)) = \frac{\Psi(I)^\top \Psi(T)}{\|\Psi(I)\| \, \|\Psi(T)\|}. \tag{1}$$

For uni-directional conversion, we use the MLLM $f$ to fuse $(I_{ref}^i, T^i)$: $T_f^i = f(I_{ref}^i, T^i, p_d) \in \mathcal{T}$ while diffusion model $g$ is used to synthesize images based on MLLM-generated descriptions $T_f^i$: $I_g^i = g(I_{ref}^i, T_f^i) \in \mathcal{I}$. To use pretrained retrieval models, we feed the generated text $T_f^i$ to retrieval model $\Psi$ to compute cosine similarity with respect to all candidate images. $s_{ij} = \cos(\Psi(T_f^i), \Psi(I_{tar}^j))$, and similarly for synthesized images $I_g^i$: $s_{ij} = \cos(\Psi(I_g^i), \Psi(I_{tar}^j))$, where $j \in \{1, \ldots, n\}$. Through this method, we reformulate CIR into two types of image retrieval tasks: text-to-image and image-to-image as in the pink box of Figure 1.

For bi-directional conversion, we additionally convert target images $I_{tar}$ to text via MLLMs given another dataset-specific prompt $q_d$: $T_f^j = f(I_{tar}^j, q_d)$ and compute the cosine similarity based on previously generated modalities, either by $s_{ij} = \cos(\Psi(T_f^i), \Psi(T_f^j))$ or $s_{ij} = \cos(\Psi(I_g^i), \Psi(T_f^j))$ for $j \in \{1, \ldots, n\}$, which reformulates the CIR tasks into text-to-text retrieval and image-to-text retrieval tasks in the green box of Figure 1, respectively. We summarize our methods in Algorithm 1. Based on the scores, we rank the candidates in descending order and return the top-$K$ candidate image IDs for performance evaluation. We compute the commonly used retrieval metrics according to the ground-truth labels and the returned top-$K$ candidates for each query on the standard official splits of datasets, and report the results.

---

**Algorithm 1** Training-Free Pseudo-Fusion (PeFuse) for Composed Image Retrieval

---

**Requirements:** Retrieval model $\Psi(\cdot)$, MLLM $f(\cdot)$, diffusion model $g(\cdot)$, prompts $p_d$ and $q_d$
**Input:** Reference image $I_{\text{ref}}$, modification text $T$, candidate image set $\mathcal{D} = \{I_{\text{tar}}^j\}_{j=1}^N$, retrieval mode $m$
**Output:** Ranked list of candidate images

1: **if** $m \in \{\text{T}\rightarrow\text{I}, \text{T}\rightarrow\text{T}\}$ **then**
2:   Generate composed text description:

$$T_f \leftarrow f(I_{\text{ref}}, T, p_d)$$

3:   **if** $m = \text{T}\rightarrow\text{I}$ **then**
4:    **for** each candidate image $I_{\text{tar}}^j \in \mathcal{D}$ **do**
5:     $s_j \leftarrow \cos\left(\Psi(T_f), \Psi(I_{\text{tar}}^j)\right)$
6:   **else if** $m = \text{T}\rightarrow\text{T}$ **then**
7:    **for** each candidate image $I_{\text{tar}}^j \in \mathcal{D}$ **do**
8:     Generate target description:
     $T_f^j \leftarrow f(I_{\text{tar}}^j, q_d)$
9:     $s_j \leftarrow \cos\left(\Psi(T_f), \Psi(T_f^j)\right)$
10: **else**
11:   Generate synthesized target image:
   $I_g \leftarrow g(I_{\text{ref}}, T_f)$
12:   **if** $m = \text{I}\rightarrow\text{I}$ **then**
13:    **for** each candidate image $I_{\text{tar}}^j \in \mathcal{D}$ **do**
14:     $s_j \leftarrow \cos\left(\Psi(I_g), \Psi(I_{\text{tar}}^j)\right)$
15:   **else if** $m = \text{I}\rightarrow\text{T}$ **then**
16:    **for** each candidate image $I_{\text{tar}}^j \in \mathcal{D}$ **do**
17:     Generate target description:
     $T_f^j \leftarrow f(I_{\text{tar}}^j, q_d)$
18:     $s_j \leftarrow \cos\left(\Psi(I_g), \Psi(T_f^j)\right)$
19: Sort candidates in descending order of $s_j$
20: **return** ranked candidate list

---

Our framework is termed pseudo-fusion as it relies on generative models to synthesize new data (images or text) from pairs of elements within a triplet, thereby achieving an implicit fusion of modalities. This approach is distinct from typical early fusion paradigms, which explicitly combine modalities into intermediate embeddings. In contrast, our method directly generates coherent and interpretable data in a target modality, preserving latent semantics throughout the process.

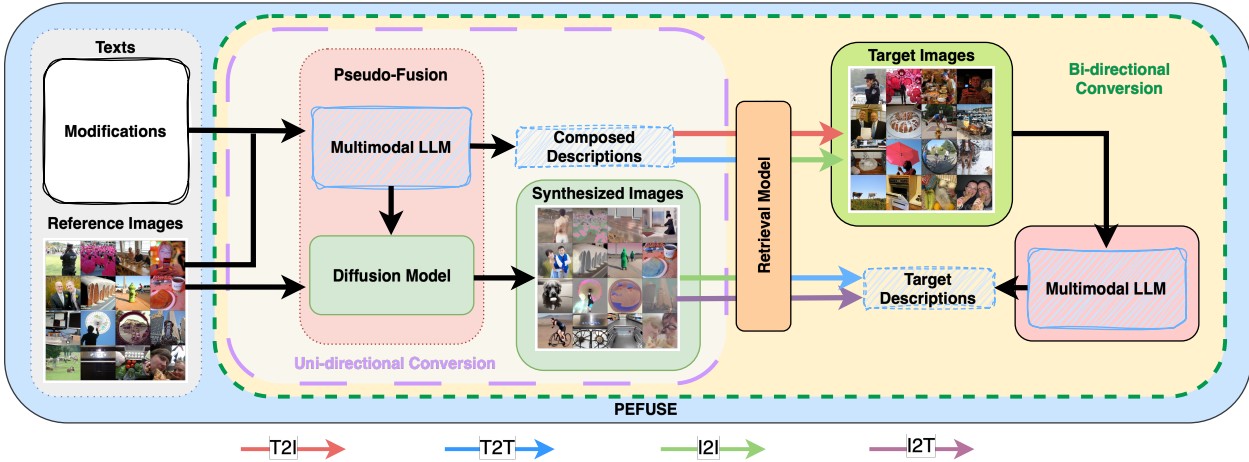

Figure 1: Training-free pseudo-fusion framework for Composed Image Retrieval. Dashed green box indicates bi-directional conversion, while the dashed purple box is uni-directional conversion. T2I: text-to-image; T2T: text-to-text; I2T: image-to-text; I2I: image-to-image. Black arrows indicate the modality conversion flow.

## 4 Experiments

We first describe the experiment setup, including the datasets and models used, and then present the experimental results for different conversion methods. Note that multi-round conversion or using multiple models can also be incorporated into our pipeline for improved performance, whereas we minimize the number of models used to investigate which pathway is the most effective when converting CIR task into a single-modality retrieval task. Specifically, our experiments use either an MLLM or a diffusion model for uni-directional conversion, while both an MLLM and a diffusion model are used for bi-directional conversion.

### 4.1 Data and Models

We employ four standard CIR benchmark datasets: Fashion-IQ (Wu et al., 2021), CIRR (Liu et al., 2021), CIRCO (Baldrati et al., 2023a), and GeneCIS (Vaze et al., 2023). Fashion-IQ is designed for interactive fashion image retrieval using natural language feedback, incorporating human-written relative captions and derived visual attributes. CIRR extends the scope to open-domain images with human-annotated modifying text, though it is known to contain a significant number of false negatives (Baldrati et al., 2023a). To mitigate this issue, CIRCO provides multiple ground-truth images per query, with all images sourced from the MS-COCO (Lin et al., 2014) dataset. In addition, GeneCIS measures models' ability to adapt to a range of similarity conditions in terms of attributes and objects. In line with standard evaluation protocols, we use the official splits of each dataset, along with the corresponding official target image candidate pools. We report recall@$K$ for Fashion-IQ, CIRR, and GeneCIS, and mean average precision (mAP@$K$) for CIRCO, reflecting their respective annotation structures. Note that predictions on the test splits of CIRR and CIRCO must be submitted to their respective evaluation servers for automatic assessment, as ground-truth labels are not publicly available.[1] In contrast, Fashion-IQ and GeneCIS provide ground-truth labels, enabling the direct computation of the corresponding evaluation metrics.

For image synthesis based on textual and visual inputs, we utilize the pretrained `sdxl-instructpix2pix` model from the `diffusers` library, which is an instruction-tuned variant of InstructPix2Pix (Brooks et al., 2023), for its strong generative performance. Text generation is handled by `Qwen2.5-VL-7B-Instruct`, an advanced instruction-tuned SOTA MLLM based on Qwen (Yang et al., 2025). We later evaluate additional MLLMs and diffusion models to compare retrieval performance and analyze computational overhead incurred by our pipeline in section 5.

---

[1] CIRR server: `https://cirr.cecs.anu.edu.au`, CIRCO server: `https://circo.micc.unifi.it`

To evaluate the effectiveness of our approach, we employ several retrieval models as feature extractors, starting with models using ViT-B/32 backbone: CLIP (ViT-B/32) (Radford et al., 2021), OpenCLIP (ViT-B/32) (Cherti et al., 2023), and SigLIP2 (Base-Patch16) (Tschannen et al., 2025). To ensure a fair comparison with methods using the same backbone and to maintain maximum consistency in our pipeline, we use SigLIP2 with ViT-B-16 instead of ViT-B-32, since the available SigLIP2 ViT-B-32 checkpoint is designed for images at a resolution of $256 \times 256$. We include SigLIP2 with ViT-B-16 as a complementary retrieval model, since ViT-B-26 has a model size comparable to the ViT-B-32 backbones used by CLIP and OpenCLIP, allowing us to control for backbone capacity as much as possible. Both CLIP and OpenCLIP adopt the softmax function in their loss functions, with OpenCLIP additionally benefiting from training on substantially larger datasets. In contrast, SigLIP2 improves semantic understanding over SigLIP (Zhai et al., 2023), which was trained with sigmoid-based contrastive loss. We investigate scaling behavior of larger model variants in subsection 4.4.

All retrieval models operate on input images resized to $224 \times 224$ pixels, normalized to the $[0, 1]$ range using model-specific normalization parameters. The diffusion model requires $768 \times 768$ pixel inputs and produces outputs at the same resolution. For consistency, all images across datasets are resized to $768 \times 768$ and normalized to $[0, 1]$ prior to diffusion processing. During image generation with the diffusion model, we use a guidance scale of 7.5, an image guidance scale of 3.0, and 30 denoising steps for performance. For text generation with the MLLM, we set the temperature to 0.1, top-$P$ to 0.9, top-$K$ to 50, and a maximum of 128 new tokens to ensure deterministic outputs with reasonable length. A sensitivity study in subsection 4.5 examines the impact of varying these hyperparameters. Our implementation uses `PyTorch` on a single NVIDIA A100 with 40 GB of memory.

### 4.2 Uni-directional conversion

As introduced in section 3, uni-directional conversion can be implemented using either MLLMs or diffusion models as pseudo-fusion methods for queries, converting CIR to single-modality image-based retrieval tasks.

Top rows (PEFUSE (T→I) and PEFUSE (I→I)) in Table 1 present the text-to-image and image-to-image retrieval results on the Fashion-IQ dataset. We mainly compare with zero-shot training-free methods: CIReVL, LDRE, WeimoCIR, and ImageScope, while training-based methods are reported in Table 8 and Appendix B. When reformulating CIR as a text-to-image retrieval task, both CLIP and OpenCLIP, which utilize the ViT-B/32 backbone, outperform CIReVL. Notably, OpenCLIP as the retriever surpasses the performance of most zero-shot methods that necessitate model training, with results comparable to, though slightly lower than LinCIR, which uses a larger ViT-L/14 backbone, and to that of WeiMoCIR.

In general, when comparing PEFUSE (T→I) and PEFUSE (I→I), text-to-image retrieval demonstrates superior performance compared to image-to-image retrieval; the sole exception is observed with the SigLIP2 model. Across all retrieval models evaluated, we note a significant inconsistency in performance: SigLIP2 yields the weakest results for text-to-image retrieval, yet achieves the strongest performance for image-to-image retrieval. This disparity underscores the substantial differences in semantic understanding capabilities across retrieval models on varying tasks, highlighting their critical and impactful role in the effectiveness of CIR systems.

The image retrieval performance on the CIRR and CIRCO datasets is further detailed in Table 2 and Table 3, respectively. Training-based baseline methods are reported in Table 9 and Table 10, with additional results provided in Appendix B. On the CIRR dataset, for the text-to-image retrieval task, the CLIP model slightly underperforms compared to other methods while being notably better on CIRR subsets. In contrast, SigLIP and OpenCLIP achieve significantly stronger performance, on both CIRR and its subsets, although they fall behind ImageScope in some cases. For the image-to-image task on CIRR, all retrieval models fall behind the established baselines, underscoring the superiority of reformulating CIR as a text-to-image rather than an image-to-image retrieval task.

A similar phenomenon is observed on the CIRCO dataset, where text-to-image retrieval outperforms most baseline methods except ImageScope, while all the baselines outperform the image-to-image conversion mode. Specifically, text-to-image retrieval using CLIP performs competitively, exceeding training-free methods such as CIReVL, though it remains behind LinCIR and HyCIR, both of which employ a larger ViT-L/14 backbone

Table 1: Performance (%) on **Fashion-IQ** validation split using different retrieval models via PᴇFᴜsᴇ. **Best** and second best highlighted. *: reproduced results; †: results from original papers. Training-based baselines are reported in Table 8 in Appendix B.

| Method | Retrieval Model | Shirt | | Dress | | Toptee | | Average | |
|---|---|---|---|---|---|---|---|---|---|
| | | R@10 | R@50 | R@10 | R@50 | R@10 | R@50 | R@10 | R@50 |
| CIReVL* | CLIP (ViT-B/32) | 18.40 | 30.82 | 14.25 | 30.45 | 18.00 | 34.33 | 16.88 | 31.87 |
| LDRE† | CLIP (ViT-B/32) | 27.38 | 46.27 | 19.97 | 41.84 | 27.07 | 48.78 | 24.81 | 45.63 |
| WeiMoCIR† | OpenCLIP (ViT-B/32) | **29.20** | **47.15** | **26.23** | **46.31** | **34.17** | **55.99** | **29.86** | **49.82** |
| ImageScope† | CLIP (ViT-B/32) | 24.29 | 37.49 | 18.00 | 35.20 | 24.99 | 41.41 | 22.42 | 38.03 |
| | CLIP | 20.62 | 37.11 | 13.99 | 32.54 | 19.93 | 39.58 | 18.18 | 36.41 |
| PᴇFᴜsᴇ (T→I) | OpenCLIP | 28.30 | 46.19 | 24.05 | 44.11 | 32.46 | 53.94 | 28.27 | 48.08 |
| | SigLIP2 | 6.49 | 13.61 | 7.32 | 17.05 | 7.23 | 16.93 | 7.01 | 15.86 |
| | CLIP | 8.97 | 17.01 | 4.95 | 13.82 | 7.71 | 16.76 | 7.21 | 15.87 |
| PᴇFᴜsᴇ (I→I) | OpenCLIP | 14.28 | 24.48 | 10.33 | 22.43 | 12.69 | 24.37 | 12.43 | 23.76 |
| | SigLIP2 | 15.00 | 26.39 | 9.25 | 20.93 | 13.77 | 25.01 | 12.67 | 24.11 |
| | CLIP | 14.33 | 25.57 | 8.82 | 20.55 | 16.01 | 29.57 | 13.06 | 25.23 |
| PᴇFᴜsᴇ (T→T) | OpenCLIP | 16.39 | 26.96 | 13.23 | 27.92 | 19.39 | 34.33 | 16.34 | 29.74 |
| | SigLIP2 | 2.53 | 5.88 | 1.72 | 5.38 | 3.05 | 7.61 | 2.43 | 6.29 |
| | CLIP | 10.67 | 20.93 | 5.16 | 16.41 | 8.94 | 20.03 | 8.26 | 19.12 |
| PᴇFᴜsᴇ (I→T) | OpenCLIP | 14.28 | 26.55 | 8.18 | 19.96 | 12.32 | 25.44 | 11.59 | 23.98 |
| | SigLIP2 | 8.61 | 17.27 | 5.59 | 15.33 | 8.94 | 19.34 | 7.72 | 17.31 |

and demand training. Notably, within the same task, using OpenCLIP and SigLIP with a ViT-B/32 backbone improves the performance significantly, and SigLIP surpasses most baseline methods by a considerable margin while being behind the top performer ImageScope. This indicates that employing a more powerful retrieval model can substantially enhance system performance. Conversely, experiments on image-to-image retrieval for CIRCO demonstrate inferior results, highlighting a need for improvement in diffusion-based conversion methods. We show results on GeneCIS in Table 4 and those of training-based methods in Table 11 in Appendix B. From Table 4, we find text-to-image still performs the best while other conversion modes are still competitive with the baselines, which further corroborates our analysis on PᴇFᴜsᴇ (T→I) and PᴇFᴜsᴇ (I→I).

The experimental results across all datasets demonstrate that the framework is effective for ZS-CIR tasks, despite its simplicity and training-free nature. Although CLIP has been widely adopted in previous studies, our results indicate that it is a suboptimal choice for CIR systems compared to OpenCLIP, emphasizing the importance of choosing appropriate retrieval models for CIR performance. We further note that methods which separately generate captions via an image captioner and then combine them with modification text using an LLM (e.g., CIReVL) can be effective for simple images, such as fashion items. However, in complex scenarios like those in CIRCO, which involve a large pool of candidate images (123K), such pipelines often fail to adequately capture the intricate interactions between reference images and textual modifications, leading to inferior retrieval performance. Consequently, employing an MLLM proves to be both sufficient and less error-prone, outperforming lengthy chained pipelines for complex image retrieval tasks. Additionally, applying methods from LDRE for MLLM-generated captions can further improve image retrieval performance.

### 4.3 Bi-directional Conversion

In addition to leveraging an MLLM to fuse the information from reference images and their corresponding modification text into textual descriptions, the same model is employed to generate descriptive captions for target images. This methodology effectively reformulates the CIR task into a text-to-text retrieval problem. Together with generated images via diffusion models, the CIR task is reframed as an image-to-text retrieval task. Previously, we reformulated the CIR as either image-to-image retrieval via a diffusion model or text-to-image retrieval via an MLLM. Now, we show the results when reformulating CIR to single-modality text retrieval tasks, i.e., PᴇFᴜsᴇ (T→T) and PᴇFᴜsᴇ (I→T), with additional conversion on target images using an MLLM. We report results in bottom rows of Table 1, Table 2, Table 3, and Table 4 respectively.

Table 2: Performance (%) on **CIRR** test split using different retrieval models via PEFUSE. **Best** and second best highlighted. *: reproduced results; †: results from original papers; −: results not available. Training-based baselines are reported in Table 9 in Appendix B.

| Method | Retrieval Model | Recall | | | | | Recallsubset | | |
|---|---|---|---|---|---|---|---|---|---|
| | | @1 | @2 | @5 | @10 | @50 | @1 | @2 | @3 |
| CIReVL* | CLIP (ViT-B/32) | 21.40 | 31.86 | 47.74 | 60.72 | 84.99 | 56.27 | 77.08 | 88.63 |
| LDRE† | CLIP (ViT-B/32) | 25.69 | — | 55.13 | 69.04 | 89.90 | 60.53 | 80.65 | 90.70 |
| WeiMoCIR† | OpenCLIP (ViT-B/32) | 26.31 | — | 57.69 | 70.36 | 91.01 | 53.35 | 75.57 | 87.76 |
| ImageScope† | CLIP (ViT-B/32) | **34.36** | — | 60.58 | 71.40 | 88.41 | **74.63** | 87.93 | 93.83 |
| PEFUSE (T→I) | CLIP | 22.58 | 32.96 | 49.81 | 63.59 | 87.47 | 63.28 | 82.27 | 91.76 |
| | OpenCLIP | 34.00 | **47.49** | **65.57** | **77.47** | **93.28** | 71.59 | **88.39** | **95.04** |
| | SigLIP2 | 30.65 | 43.13 | 60.60 | 72.43 | 91.35 | 70.00 | 86.48 | 93.64 |
| PEFUSE (I→I) | CLIP | 2.77 | 9.78 | 23.06 | 35.42 | 64.58 | 29.90 | 52.19 | 70.96 |
| | OpenCLIP | 3.28 | 11.64 | 27.76 | 41.45 | 71.37 | 31.01 | 53.83 | 72.12 |
| | SigLIP2 | 3.78 | 12.72 | 28.39 | 42.12 | 70.07 | 32.43 | 54.15 | 71.35 |
| PEFUSE (T→T) | CLIP | 21.81 | 31.13 | 45.06 | 56.72 | 78.51 | 59.47 | 79.66 | 90.48 |
| | OpenCLIP | 30.65 | 42.96 | 58.99 | 70.68 | 89.25 | 69.45 | 86.00 | 93.81 |
| | SigLIP2 | 16.07 | 24.19 | 37.40 | 47.81 | 68.82 | 52.77 | 72.39 | 84.65 |
| PEFUSE (I→T) | CLIP | 6.80 | 13.76 | 28.27 | 42.12 | 72.27 | 36.12 | 57.83 | 74.80 |
| | OpenCLIP | 7.40 | 15.86 | 31.71 | 45.16 | 74.29 | 36.10 | 58.72 | 75.47 |
| | SigLIP2 | 7.52 | 14.12 | 26.82 | 39.13 | 67.49 | 34.41 | 56.63 | 74.41 |

Table 3: Performance (%) on **CIRCO** test split using different retrieval models via PEFUSE. **Best** and second best highlighted. *: reproduced results; †: results from original papers. Training-based baselines are reported in Table 10 in Appendix B.

| Method | Retrieval Model | mAP@5 | mAP@10 | mAP@25 | mAP@50 |
|---|---|---|---|---|---|
| CIReVL* | CLIP (ViT-B/32) | 10.36 | 10.70 | 11.88 | 12.47 |
| LDRE† | CLIP (ViT-B/32) | 17.96 | 18.32 | 20.21 | 21.11 |
| ImageScope† | CLIP (ViT-B/32) | **22.36** | **22.19** | **23.03** | **23.83** |
| PEFUSE (T→I) | CLIP | 11.12 | 11.47 | 12.86 | 13.61 |
| | OpenCLIP | 16.89 | 17.56 | 19.14 | 20.16 |
| | SigLIP2 | 18.53 | 19.68 | 21.58 | 22.62 |
| PEFUSE (I→I) | CLIP | 2.39 | 2.57 | 3.08 | 3.37 |
| | OpenCLIP | 3.03 | 3.49 | 4.10 | 4.44 |
| | SigLIP2 | 4.06 | 4.63 | 5.47 | 5.94 |
| PEFUSE (T→T) | CLIP | 7.71 | 7.97 | 8.76 | 9.26 |
| | OpenCLIP | 11.52 | 11.86 | 13.07 | 13.71 |
| | SigLIP2 | 6.68 | 6.56 | 7.14 | 7.48 |
| PEFUSE (I→T) | CLIP | 3.89 | 4.35 | 4.88 | 5.29 |
| | OpenCLIP | 4.10 | 4.62 | 5.26 | 5.74 |
| | SigLIP2 | 3.65 | 3.84 | 4.40 | 4.78 |

We observe that reformulating CIR as a text-to-text retrieval task generally yields stronger performance compared to image-to-text retrieval. Furthermore, both text-to-text and image-to-text retrieval underperform relative to text-to-image retrieval. However, image-to-text retrieval generally surpasses image-to-image performance. Overall, using diffusion-generated images as queries to retrieve either original target images or MLLM-generated textual descriptions, i.e., PEFUSE (I→I) and PEFUSE (I→T), consistently underperforms methods that use MLLM-generated textual descriptions as queries, namely PEFUSE (T→I) and PEFUSE (T→T). One possible explanation is that images synthesized by diffusion models often contain artifacts that distinguish them from natural images. See Appendix D for generated examples. These artifacts can be captured by the image encoder of the retrieval model, thereby increasing the modality gap and degrading retrieval performance, whereas text generated by MLLMs retains more semantically meaningful information. Furthermore, as reported in Mistretta et al. (2025), models trained with cross-modal contrastive objectives generally perform better on cross-modal retrieval tasks than on intra-modal retrieval tasks (e.g., image-to-image or text-to-text retrieval). This is because cross-modal contrastive learning explicitly aligns representations across modalities but does not impose strong constraints on relationships within the same modality. Moreover, incorporating additional generative models into the retrieval pipeline may introduce

Table 4: Performance (%) comparison on **GeneCIS** test split using different retrieval models via PeFuse. **Best** and second best highlighted. †: results from original papers. Training-based baselines are reported in Table 11 in Appendix B.

| Method | Retrieval Model | Focus Attribute | | | Change Attribute | | | Focus Object | | | Change Object | | | Average |
|---|---|---|---|---|---|---|---|---|---|---|---|---|---|---|
| | | R@1 | R@2 | R@3 | R@1 | R@2 | R@3 | R@1 | R@2 | R@3 | R@1 | R@2 | R@3 | R@1 |
| CIReVL† | CLIP (ViT-B/32) | 17.90 | 29.40 | 40.40 | **14.80** | 25.80 | 35.80 | 14.60 | 24.30 | 33.30 | 16.10 | 27.80 | 37.60 | 15.90 |
| PeFuse (T→I) | CLIP | 18.20 | 31.15 | 41.85 | 14.54 | 25.76 | 35.89 | 13.67 | 25.26 | 34.90 | 17.40 | 29.03 | 38.67 | 15.95 |
| | OpenCLIP | 19.55 | 31.95 | 43.70 | 14.73 | 26.28 | 36.51 | 17.96 | 28.27 | 38.06 | 16.73 | 30.97 | 41.33 | 17.22 |
| | SigLIP2 | 17.00 | 29.35 | 39.65 | 14.63 | 26.70 | 36.17 | 18.27 | 29.34 | 38.27 | 19.59 | 31.58 | 42.24 | 17.37 |
| PeFuse (I→I) | CLIP | 15.90 | 27.15 | 37.50 | 11.03 | 20.36 | 27.60 | 9.18 | 18.88 | 27.35 | 8.16 | 17.60 | 27.65 | 11.07 |
| | OpenCLIP | 17.60 | 30.65 | 41.70 | 11.22 | 21.07 | 29.36 | 10.41 | 19.90 | 28.42 | 9.08 | 18.83 | 28.06 | 12.08 |
| | SigLIP2 | 17.95 | 29.15 | 40.00 | 11.17 | 21.69 | 30.21 | 10.20 | 19.49 | 27.91 | 9.90 | 20.71 | 30.71 | 12.31 |
| PeFuse (T→T) | CLIP | 16.55 | 28.80 | 39.65 | 10.18 | 20.45 | 29.02 | 14.85 | 24.44 | 33.27 | 12.30 | 23.98 | 32.60 | 13.47 |
| | OpenCLIP | 16.60 | 30.90 | 41.95 | 12.26 | 22.44 | 32.81 | 16.53 | 27.35 | 35.87 | 16.17 | 27.91 | 37.96 | 15.39 |
| | SigLIP2 | 14.60 | 25.75 | 35.65 | 9.42 | 17.95 | 25.95 | 11.84 | 20.00 | 28.37 | 11.02 | 19.64 | 29.44 | 11.72 |
| PeFuse (I→T) | CLIP | 17.45 | 30.70 | 41.00 | 10.94 | 21.40 | 30.45 | 10.97 | 19.74 | 28.16 | 11.58 | 20.82 | 29.64 | 12.74 |
| | OpenCLIP | 17.45 | 30.45 | 41.75 | 10.84 | 21.40 | 31.20 | 11.84 | 21.89 | 30.77 | 11.22 | 21.07 | 30.26 | 12.84 |
| | SigLIP2 | 17.40 | 29.60 | 39.80 | 9.90 | 20.69 | 31.11 | 11.22 | 20.61 | 28.78 | 10.51 | 19.80 | 29.54 | 12.26 |

Table 5: Scaling law when using different backbones for OpenCLIP for text-to-image retrieval on each benchmark.

| Backbone | Fashion-IQ | | CIRR | | | | | CIRCO | | | | GeneCIS |
|---|---|---|---|---|---|---|---|---|---|---|---|---|
| | R@10 | R@50 | R@1 | R@2 | R@5 | R@10 | R@50 | mAP@5 | mAP@10 | mAP@25 | mAP@50 | R@1 |
| ViT-L/14 | 29.49 | 48.37 | 36.17 | 50.07 | 67.13 | 78.72 | 94.17 | 21.69 | 22.99 | 25.07 | 26.14 | 16.85 |
| ViT-H/14 | 30.40 | 50.09 | 38.55 | 52.02 | 69.49 | 80.29 | 94.29 | 23.78 | 24.78 | 27.10 | 28.24 | 17.50 |
| ViT-g/14 | 30.15 | 50.12 | 38.41 | 52.15 | 70.15 | 80.27 | 94.58 | 22.62 | 23.93 | 26.39 | 27.42 | 17.39 |
| ViT-bigG/14 | 30.44 | 49.49 | 40.41 | 54.63 | 71.13 | 81.06 | 94.89 | 25.03 | 26.63 | 29.17 | 30.28 | 17.99 |

cumulative errors at multiple stages, which can further contribute to performance degradation. These findings further demonstrate that reformulating CIR as a text-to-image retrieval task is generally more effective than other conversion strategies under the same settings when employing CLIP-style retrieval models.

### 4.4 Scaling Law

Having evaluated performance of different conversion strategies using retrieval models with a ViT-B/32 backbone in subsection 4.2 and subsection 4.3, a subsequent question arises regarding the potential benefits of larger backbone architectures. To investigate this, we assess the performance of OpenCLIP, which is selected for its superior overall performance among the three retrieval models, using ViT backbones of varying sizes across all datasets for text-to-image retrieval tasks due to their superior performance compared to other conversion modes. The overall results are presented in Table 5, with detailed category-specific results for Fashion-IQ and CIRR subsets provided in Table 12 in Appendix C, and category results of GeneCIS can be found in Table 13 in Appendix C.

As shown in Table 5, we observe a general trend of improving performance for the text-to-image retrieval task as the backbone size increases, although performance fluctuates across specific model sizes. Notably, on Fashion-IQ, Recall@10 decreases and Recall@50 saturates when using the ViT-g/14 backbone. Performance on CIRR also improves consistently with model scale, with the exception of a slight decrease in Recall@1 and Recall@10 for ViT-g/14. A similar performance drop with ViT-g/14 is observed on the CIRCO dataset. A similar trend holds for GeneCIS. Furthermore, this trend of scaling benefits extends beyond text-to-image retrieval; larger models consistently achieve superior performance on image-to-image, text-to-text, and image-to-text retrieval tasks as well when using our proposed methods to reformulate CIR tasks.

### 4.5 Sensitivity Analysis

We study how the model performance can be affected by varying values for hyperparameters of the MLLM and the diffusion model used. We quantify the relationship between CIR performance and these hyperparameters via experimental studies.

### 4.5.1 Multimodal Large Language Models

For MLLMs, temperature controls the determinism of the LLM when generating tokens, Top-$P$ is the cumulative probability threshold used when selecting tokens up to probability $P$, and Top-$K$ sampling limits the model to consider only the k most likely tokens at each step. For consistency with previous experiments, we use 0.1 for temperature, 0.9 for top-$P$, 50 for top-$K$ as the base combination and only alter one parameter while the others are fixed. For example, when we investigate how temperature would impact the retrieval performance, we fix top-$P$ to 0.9 and top-$K$ to 50, and change values for temperature in the range (0, 1). We use the average of mAP@$K$ for the y-axis, where $K \in \{1, 5, 10, 25, 50\}$.

We investigate the performance of different values of hyperparameters on CIRCO validation dataset (due to its manageable size and diverse nature of images) using OpenCLIP as the retrieval model in Figure 2. From Figure 2, we observe that retrieval performance is more strongly affected by temperature instead of top-$P$ or top-$K$. At higher temperatures, the model will produce more diverse tokens, which hurts retrieval performance when matching with images, whereas the performance of top-$P$ and top-$K$ is very stable overall. Overall, performance under the same set of hyperparameters shows slight differences, especially for top-$P$ and top-$K$. The figure indicates that a lower temperature and a moderate top-$P$ with higher top-$K$ would result in better retrieval performance.

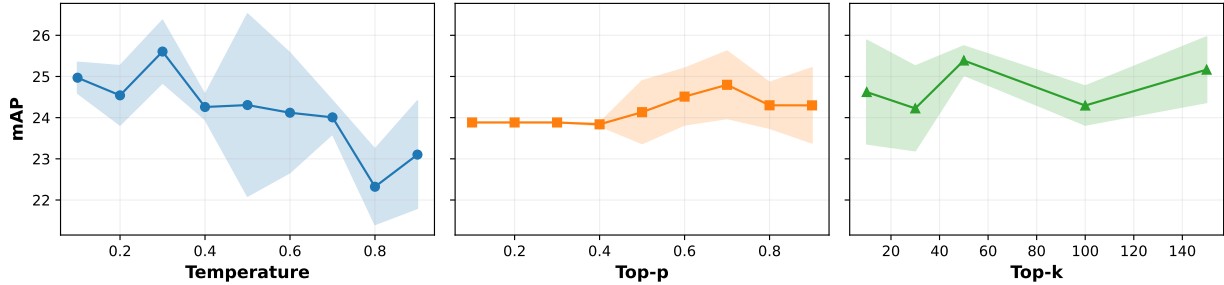

Figure 2: CIR performance (%) of PEFUSE (T→I) via the MLLM under varying values for hyperparameters on CIRCO validation split using OpenCLIP with ViT-B/32 backbone to perform text-to-image task for 3 runs. Shadowed regions indicate standard deviation.

### 4.5.2 Diffusion Models

In the inference process of diffusion models, a strong correlation exists between key hyperparameters and the properties of the synthesized output. Specifically, a larger number of inference steps correlates strongly with enhanced photorealism of the generated images. Furthermore, increasing the image guidance scale elevates the fidelity of the output to a given reference image. Conversely, a higher text guidance scale promotes stricter adherence to the input text prompt, often at the expense of output diversity. We use 7.5 for guidance scale, 3.0 for image guidance scale, and 30 for inference steps as the base combination and only change one hyperparameter and fix the rest.

We show the retrieval performance when using different values for hyperparameters in diffusion models on the CIRCO validation dataset *with* and *without* MLLMs in Figure 3. From Figure 3, we observe that using MLLM-generated descriptions to generate images improves the retrieval performance for all three hyperparameters, which validates the importance of using MLLM-generated descriptions as prompts instead of the original captions from datasets. We also show qualitative results with and without MLLMs for diffusion models in Appendix D. From the synthesized images, we observe more distinguishable artifacts when directly using raw modifications from CIRCO dataset.

The performance gap between using and not using MLLMs increases with stronger guidance scales and more inference steps while the performance gap is closing with larger values for image guidance scale. Across three hyperparameters, image guidance scale has the most significant impact on retrieval performance, and higher values cause much worse performance. With larger image guidance scale, the generated images would be

more similar to the original input images than to the intended target images, thus deviating from target images and leading to worse performance. This indicates the importance of low values for image guidance scale to achieve good performance when reformulating CIR task to image-to-image task.

We also noticed that retrieval performance is slightly increasing with a varying number of inference steps, as more inference steps produce more photorealistic images but requires more time. For time efficiency, a moderate number of inference steps should be sufficient, as indicated from the figure. Generally speaking, the retrieval performance when employing diffusion models via PEFUSE (I→I) is consistent across different runs.

Finally, we acknowledge that carefully tuning values for hyperparameters is labor-intensive, as different datasets and models might perform differently for the same setting of hyperparameters.

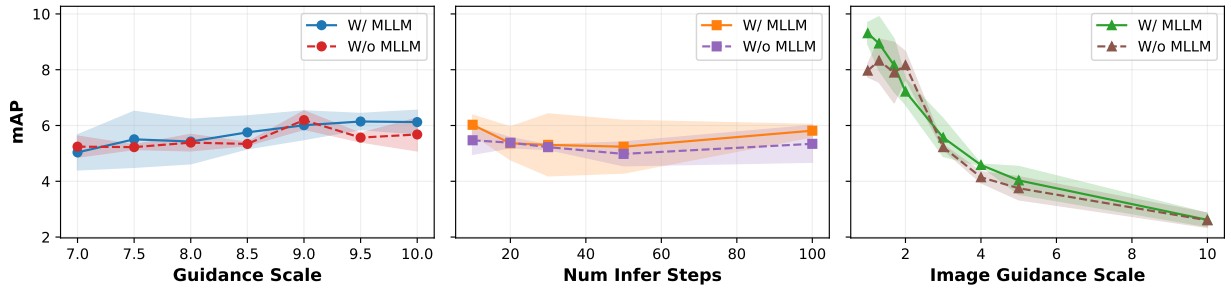

Figure 3: CIR performance (%) of PEFUSE (I→I) via diffusion models *with* and *without* using composed descriptions from the MLLM on CIRCO validation split using OpenCLIP with ViT-B/32 for 3 runs. Shadowed regions indicate standard deviation.

## 5 Ablation and Latency Analysis

To confirm how the MLLMs and diffusion models impact or contribute to downstream retrieval performance, we replace the `Qwen2.5-VL-7B` and `SDXL-InstructPix2Pix` with other pre-trained models and measure the metric of interest, which serves as an ablation of the components in our pipeline. To quantify the latency incurred and obtain a clearer picture of the time efficiency of our pipeline, we also analyze the computational overhead of the proposed framework for the CIR task. We focus on PEFUSE (T→I) and PEFUSE (I→I), as they involve both MLLMs and diffusion models. The former leverages MLLMs to compose target descriptions, while the latter employs diffusion models, either guided by MLLM-generated descriptions or by raw modifications, to generate target images. It is worth noting that both settings involve image-retrieval-based uni-directional conversion, without converting the generated target images into text.

For consistency, we use OpenCLIP (ViT-B-32) on CIRCO validation split for both tasks across experiments. Specifically, we additionally employ the `Qwen2.5-VL-3B` to examine the latency incurred by MLLM scaling, and the similarly sized `LLaVA-1.5-7B-hf` (Liu et al., 2024) to investigate the impact of different training strategies. Notably, `Qwen2.5-VL-7B` is trained with large-scale joint vision-language pretraining, whereas `LLaVA-1.5-7B-hf` aligns visual features with those of a frozen LLM. In addition, we include `SDXL-Turbo` (Sauer et al., 2024) with a comparable parameter scale to `SDXL-InstructPix2Pix`, but support faster inference, thereby illustrating the trade-off between inference speed and output quality.

We report the GPU memory consumption of each generative model considered during inference under `torch.bfloat16`, the average inference time per query, and the entire pipeline time, defined as the average end-to-end processing time per sample, including: dataset loading, model loading, data generation, feature extraction, and performance evaluation. To isolate the effect of different components on retrieval performance, we also consider three retrieval settings as baseline evaluations: (1) reference image-to-target image, (2) modification text-to-target image, and (3) reference image plus modification text-to-target image retrieval. These settings are deterministic and performed via representation extraction using OpenCLIP. For all three settings, we directly apply retrieval models to extract respective representations for both image and text modalities, and compute the cosine similarity between them. Particularly, representations

of reference images and those of corresponding modifications are added and normalized for reference image plus modification text-to-target image retrieval. Note that, for the baseline evaluations, the pipeline time directly reflects retrieval time, since no generation step is involved. For PeFuse (T→I) with multiple MLLMs, caption-generation cost is captured by the inference time. For PeFuse (I→I) with diffusion models, the inference time represents the per-query image-generation time for both `SDXL-InstructPix2Pix` and `SDXL-Turbo`. When diffusion models are chained with `Qwen2.5-VL-7B`, the reported inference time includes both stages: first, generating the target-image description with the MLLM, and second, using that description as the refined prompt together with the reference image to generate an image with a diffusion model. We adopt the same metric (the average mAP across $\{1, 5, 10, 25, 50\}$), a batch size of 8 (for limited memory), and the same MLLM and diffusion sampling hyperparameters as in subsection 4.5. For `SDXL-Turbo` we use 60 denoising steps and an image guidance scale of 0.5, resulting in $60 \times 0.5 = 30$ effective inference steps.[2]

We show the compute information in Table 6. In Table 6, the performance of all baseline evaluation exhibits zero standard deviation across runs due to deterministic representations from OpenCLIP on the split. From the table, we observe that using textual modifications to retrieve target images yields slightly better performance than using the reference image alone. The performance of the combined query falls between these two approaches. Moreover, representation extraction is computationally efficient for all baseline evaluations. Since none of these processes employ generative models, memory consumption and generative inference time are not reported. For PeFuse (T→I), we observe that `Qwen2.5-VL-3B` has the best performance among three MLLMs, while `LLaVA-1.5-7B-hf` performs the worst, which stresses the important role of appropriate MLLMs when converting multimodal queries to text. Another valuable observation is that `Qwen2.5-VL-3B` surpasses `Qwen2.5-VL-7B` by 0.18% for average mAP while being slower by 0.28 seconds per sample at inference time. On the other hand, we utilize `Qwen2.5-VL-7B` for PeFuse (I→I), for consistency. In Table 6, `SDXL-Turbo` is more than 3× faster and outperforms `SDXL-InstructPix2Pix` by 2.18% when using raw modifications. When employing generated descriptions by `Qwen2.5-VL-7B`, a similar phenomenon is observed. Interestingly, using the MLLM produces performance gains for `SDXL-InstructPix2Pix` (↑ 0.11%) whereas the performance of `SDXL-Turbo` drops by 0.22% when conditioned on generated target descriptions. When using PeFuse (T→I), the retrieval performance consistently surpasses that of all baselines that do not incorporate generative MLLMs. Similarly, when using PeFuse (I→I), all diffusion-based models improve retrieval performance compared with using the reference image alone for target image retrieval. These results collectively demonstrate the effectiveness of our approach in leveraging generative models to enhance retrieval performance. Overall, the choice of generative model affects downstream performance when using our method; however, the impact is generally marginal and remains consistent across multiple random seeds when a specific model is selected.

Table 6: Ablation and compute latency on the CIRCO validation split over 3 runs. For baseline performance, all retrieval tasks are deterministic, thus no standard deviation is reported; no generative models are involved, therefore no memory consumption is measured. For PeFuse conversion modes, generative inference time and the entire pipeline time are reported *per query*, with mean and standard deviation. "SDXL-Instr." denotes `SDXL-InstructPix2Pix`.

| Method | Generative Models | Memory (MB) | Inference Time (s) | Pipeline Time (s) | Avg mAP (%) |
|---|---|---|---|---|---|
| Modification→Target | – | – | – | 0.48 (±0.02) | 7.36 |
| Reference→Target | – | – | – | 0.48 (±0.02) | 3.48 |
| Reference+modification→Target | – | – | – | 0.49 (±0.02) | 6.61 |
| PeFuse (T→I) | Qwen2.5-VL-3B | 7162 | 0.69 (± 0.03) | 1.70 (± 0.11) | 25.15 (± 0.80) |
| | Qwen2.5-VL-7B | 15818 | 0.41 (± 0.03) | 1.70 (± 0.14) | 24.97 (± 0.37) |
| | LLaVA-1.5-7B-hf | 13472 | 0.70 (± 0.04) | 3.01 (± 0.10) | 17.11 (± 1.09) |
| PeFuse (I→I) | SDXL-Instr. | 6725 | 3.95 (± 0.02) | 6.24 (± 0.05) | 5.22 (± 0.08) |
| | SDXL-Turbo | 6725 | 1.25 (± 0.02) | 3.07 (± 0.27) | 7.40 (± 0.56) |
| | Qwen2.5-VL-7B + SDXL-Instr. | 15818 + 6725 | 4.40 (± 0.03) | 6.33 (± 0.23) | 5.33 (± 0.70) |
| | Qwen2.5-VL-7B + SDXL-Turbo | 15818 + 6725 | 1.69 (± 0.05) | 3.49 (± 0.33) | 7.18 (± 0.41) |

---

[2]https://huggingface.co/stabilityai/sdxl-turbo

## 6    Model Deployment

Our proposed framework leverages MLLMs and diffusion models, making efficient real-time deployment critical for practical applications. To ensure low-latency retrieval, we preprocess the entire candidate image pool by indexing their visual representations in a vector database using the `FAISS` (Johnson et al., 2019) library. MLLMs or diffusion models can be used offline to generate new data based on multimodal queries, and then store the query representation using the same strategy. In this way, decoupling the generative process from the retrieval step can balance performance and efficiency, enabling scalable real-time operation.

## 7    Limitations

While our method demonstrates promising performance via multiple conversion strategies for CIR tasks, and is highly compatible with comparable techniques like LDRE and ImageScope, it is subject to several challenges inherent to its design, which present exciting opportunities for future research at the same time. First, the integration of diffusion models and MLLMs for the pseudo-fusion of modalities can lead to concept drift issues, which often propagate errors through the pipeline. Consequently, chaining multiple models can be error-prone and presents a significant challenge for optimizing CIR performance via the entire pipeline. Second, our framework considers non-preprocessed user-provided text modifiers as inputs to MLLMs. We posit that performance could be enhanced through advanced text paraphrasing, structured formatting, or more sophisticated prompt engineering strategies. However, such techniques must be carefully designed to mitigate the inherent risk of LLM hallucinations. Third, diffusion models and MLLMs are highly sensitive to their hyperparameters; nevertheless, tuning these hyperparameters is a labor-intensive process that may lack generalization across diverse datasets. Ultimately, generating image captions via MLLMs or new images via diffusion models can be computationally expensive at web scale, which is a common limitation for the LLM/MLLM-based methods (Karthik et al., 2024; Yang et al., 2024; Luo et al., 2025; Wu et al., 2025).

Based on these limitations, we identify several promising directions for future work. Efforts could focus on developing robust integration techniques to minimize error propagation in multi-model pipelines. Furthermore, considering more complex scenarios remains a compelling long-term goal. For example, compositions involving multiple input images, longer text narratives, or additional modalities such as video, are worth investigating. Another promising avenue is to explore iterative, multi-round fusion of images and text to generate progressively more refined and more accurate descriptions. Finally, such conversion paradigms inherently introduce some latency, therefore, to massively deploy interpretable CIR systems as such, more sophisticated and lightweight models should be developed.

## 8    Conclusion

This work represents the first systematic investigation and benchmarking of training-free pseudo-fusion for both uni-directional and bi-directional conversion within ZS-CIR. We empirically quantified the relationship between CIR performance and key hyperparameters of modern generative models, and conducted a ablation and latency analysis when using various MLLMs and diffusion models. Our results demonstrate that CIR tasks can be effectively reformulated by converting heterogeneous modalities into a single, unified modality. This approach enables the use of standard single-query retrieval systems, either intra-modal or cross-modal, leveraging existing high-performance pre-trained models in a plug-and-play manner, without having to train new modules. Furthermore, our analysis establishes that reformulating the CIR task as a text-to-image retrieval task is a more effective strategy compared with other conversion modes, emphasizing the importance of choosing appropriate MLLMs and diffusion models when converting multimodal queries. The competitive performance of generative models in this pseudo-fusion role underscores their potential as a powerful tool for modality unification and points to a promising future for generative, model-based fusion methods in multimodal machine learning.

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

## A  Datasets

Table 7 shows the details of each dataset we used in our experiments. Due to broken links in the Fashion-IQ dataset, we are missing some images comparing to the original dataset: shirt missing 98, dress 158, and toptee 97. CIRR, CIRCO, and GeneCIS datasets have all images available.

Table 7: Public benchmarks used in our experiments. We use the validation split for Fashion-IQ and test splits for CIRCO, CIRR, and GeneCIS.

| Dataset | # of Queries | # of Candidates |
|---|---|---|
| Fashion-IQ (Shirt) | 1940 | 6181 |
| Fashion-IQ (Dress) | 1859 | 3648 |
| Fashion-IQ (Toptee) | 1867 | 5261 |
| CIRCO | 800 | 123403 |
| CIRR | 4148 | 2315 |
| GeneCIS (Focus Attr) | 2000 | 20000 |
| GeneCIS (Change Attr) | 2112 | 31680 |
| GeneCIS (Focus Obj) | 1960 | 29400 |
| GeneCIS (Change Obj) | 1960 | 29400 |

## B  Training-based Methods

To gain further insights, we provide the baseline performance of training-based methods on all datasets in Table 8, Table 10, Table 9, and Table 11, respectively. Nonetheless, we mainly focus on training-free methods in this paper.

## C  Scaling law on Categories of Fashion-IQ, CIRR subsets, and GeneCIS categories

We report results of scaling law for each category of Fashion-IQ dataset and the subsets of CIRR in Table 12 and those of GeneCIS in Table 13.

## D  Qualitative Results Using MLLM for Diffusion Models

We show the superiority of generated images based on MLLM-generated text over raw captions for CIRCO validation split in Figure 4 and Figure 5. It can be observed that using raw captions to generate images

Table 8: Performance (%) of training-based methods on **Fashion-IQ** validation split. **Best** and second best highlighted. *: reproduced results; †: results from original papers.

| Method | Retrieval Model | Shirt | | Dress | | Toptee | | Average | |
|---|---|---|---|---|---|---|---|---|---|
| | | R@10 | R@50 | R@10 | R@50 | R@10 | R@50 | R@10 | R@50 |
| Pic2Word† | CLIP (ViT-L/14) | 26.20 | 43.60 | 20.00 | 40.20 | 27.90 | 47.40 | 24.70 | 43.70 |
| SEARLE-OTI* | CLIP (ViT-B/32) | 24.43 | 41.39 | 19.85 | 40.72 | 24.85 | 45.47 | 23.05 | 42.53 |
| SEARLE* | CLIP (ViT-B/32) | 24.85 | 41.60 | 19.37 | 39.21 | 25.12 | 46.22 | 23.11 | 42.34 |
| CoLLM† | CLIP (ViT-B/32) | 24.90 | 45.10 | 22.90 | 43.80 | 26.40 | 46.80 | 24.80 | 45.20 |
| LinCIR* | CLIP (ViT-L/14) | 29.69 | 48.14 | 22.32 | 45.13 | 30.85 | 52.01 | 27.62 | 48.43 |
| SEARLE+CIG-XL turbo† | CLIP (ViT-B/32) | 24.73 | 41.46 | 18.94 | 39.66 | 25.50 | 46.66 | 23.06 | 42.59 |
| HyCIR† | CLIP (ViT-L/14) | 27.62 | 44.94 | 19.98 | 40.80 | 28.14 | 47.67 | 25.25 | 44.47 |

Table 9: Performance (%) of training-based methods on **CIRR** test split. *: reproduced results; †: results from original papers; −: results not available.

| Method | Retrieval Model | Recall | | | | | Recall$_{subset}$ | | |
|---|---|---|---|---|---|---|---|---|---|
| | | @1 | @2 | @5 | @10 | @50 | @1 | @2 | @3 |
| Pic2Word† | CLIP (ViT-L/14) | 23.90 | — | 51.70 | 65.30 | 87.80 | 53.76 | 74.46 | 87.08 |
| SEARLE-OTI* | CLIP (ViT-B/32) | 23.18 | 34.72 | 52.31 | 66.00 | 89.21 | 52.02 | 74.43 | 86.75 |
| SEARLE* | CLIP (ViT-B/32) | 23.33 | 34.89 | 52.89 | 66.99 | 89.81 | 53.90 | 76.19 | 87.76 |
| CoLLM† | CLIP (ViT-B/32) | 28.60 | — | — | 71.80 | 92.70 | — | — | — |
| LinCIR* | CLIP (ViT-L/14) | 25.04 | 36.22 | 53.78 | 67.18 | 88.75 | 56.53 | 76.82 | 88.70 |
| SEARLE+CIG-XL turbo† | CLIP (ViT-B/32) | 25.54 | — | 55.01 | 68.24 | 90.72 | 57.52 | 78.36 | 89.35 |
| HyCIR† | CLIP (ViT-L/14) | 25.08 | — | 53.49 | 67.03 | 89.85 | 53.83 | 75.06 | 87.18 |

Table 10: Performance (%) of training-based methods on **CIRCO** test split. *: reproduced results; †: results from original papers; −: results not available.

| Method | Retrieval Model | mAP@5 | mAP@10 | mAP@25 | mAP@50 |
|---|---|---|---|---|---|
| Pic2Word‡ | CLIP (ViT-L/14) | 8.72 | 9.51 | 10.64 | 11.29 |
| SEARLE-OTI* | CLIP (ViT-B/32) | 7.29 | 7.99 | 9.21 | 9.85 |
| SEARLE* | CLIP (ViT-B/32) | 9.38 | 9.95 | 11.13 | 11.85 |
| CoLLM† | CLIP (ViT-B/32) | 12.90 | 13.20 | — | 15.00 |
| LinCIR* | CLIP (ViT-L/14) | 12.33 | 13.13 | 14.56 | 15.46 |
| SEARLE+CIG-XL turbo† | CLIP (ViT-B/32) | 10.45 | 11.02 | 12.34 | 13.00 |
| HyCIR† | CLIP (ViT-L/14) | 14.12 | 15.02 | 16.72 | 17.56 |

Table 11: Performance (%) of training-based methods on **GeneCIS** test split.

| Method | Retrival Model | Focus Attribute | | | Change Attribute | | | Focus Object | | | Change Object | | | Average |
|---|---|---|---|---|---|---|---|---|---|---|---|---|---|---|
| | | R@1 | R@2 | R@3 | R@1 | R@2 | R@3 | R@1 | R@2 | R@3 | R@1 | R@2 | R@3 | R@1 |
| Pic2Word† | CLIP (ViT-L/14) | 15.65 | 28.16 | 38.65 | 13.87 | 24.67 | 33.05 | 8.42 | 18.01 | 25.77 | 6.68 | 15.05 | 24.03 | 11.16 |
| SEARLE† | CLIP (ViT-B/32) | 18.90 | 30.60 | 41.20 | 13.00 | 23.80 | 33.70 | 12.20 | 23.00 | 33.30 | 13.60 | 23.80 | 33.30 | 14.40 |
| LinCIR† | CLIP (ViT-L/14) | 16.90 | 29.95 | 41.45 | 16.19 | 27.98 | 36.84 | 8.27 | 17.40 | 26.22 | 7.40 | 15.71 | 25.00 | 12.19 |
| LinCIR+CIG-XL turbo† | CLIP (ViT-L/14) | 16.80 | 29.70 | 40.90 | 15.91 | 28.88 | 37.45 | 8.37 | 17.35 | 25.10 | 7.86 | 15.46 | 24.29 | 12.24 |

Table 12: Scaling law on each category of **Fashion-IQ** and **CIRR** subsets using OpenCLIP for the text-to-image retrieval task.

| Backbone | Shirt | | Dress | | Toptee | | CIRR | | |
|---|---|---|---|---|---|---|---|---|---|
| | R@10 | R@50 | R@10 | R@50 | R@10 | R@50 | R$_{subset}$@1 | R$_{subset}$@2 | R$_{subset}$@3 |
| ViT-L/14 | 29.54 | 47.01 | 25.23 | 43.73 | 33.69 | 54.37 | 73.49 | 88.82 | 95.08 |
| ViT-H/14 | 30.41 | 47.22 | 26.90 | 47.71 | 33.90 | 55.33 | 74.41 | 89.23 | 95.33 |
| ViT-g/14 | 30.52 | 48.87 | 25.12 | 46.05 | 34.82 | 55.44 | 74.15 | 89.57 | 95.45 |
| ViT-bigG/14 | 31.39 | 47.32 | 25.12 | 45.56 | 34.82 | 55.60 | 75.90 | 89.37 | 95.59 |

Table 13: Scaling law on each category of **GeneCIS** using OpenCLIP for text-to-image task.

| Backbone | Focus Attribute | | | Change Attribute | | | Focus Object | | | Change Object | | |
|---|---|---|---|---|---|---|---|---|---|---|---|---|
| | R@1 | R@2 | R@3 | R@1 | R@2 | R@3 | R@1 | R@2 | R@3 | R@1 | R@2 | R@3 |
| ViT-L/14 | 17.85 | 30.30 | 41.90 | 14.25 | 27.27 | 37.12 | 16.63 | 27.50 | 38.21 | 18.67 | 31.02 | 40.61 |
| ViT-H/14 | 19.10 | 31.35 | 43.50 | 15.58 | 27.18 | 37.97 | 17.50 | 28.16 | 37.09 | 17.81 | 29.59 | 39.54 |
| ViT-g/14 | 19.25 | 32.05 | 42.10 | 15.77 | 27.94 | 37.83 | 17.35 | 27.50 | 37.24 | 17.19 | 29.69 | 39.39 |
| ViT-bigG/14 | 19.00 | 31.15 | 42.95 | 16.57 | 28.79 | 39.44 | 17.55 | 27.91 | 37.45 | 18.83 | 30.77 | 40.87 |

incurs more distinguishable artifacts for both diffusion models, and SDXL-Turbo-generated images have more artifacts than those of SDXL-InstructPix2Pix.

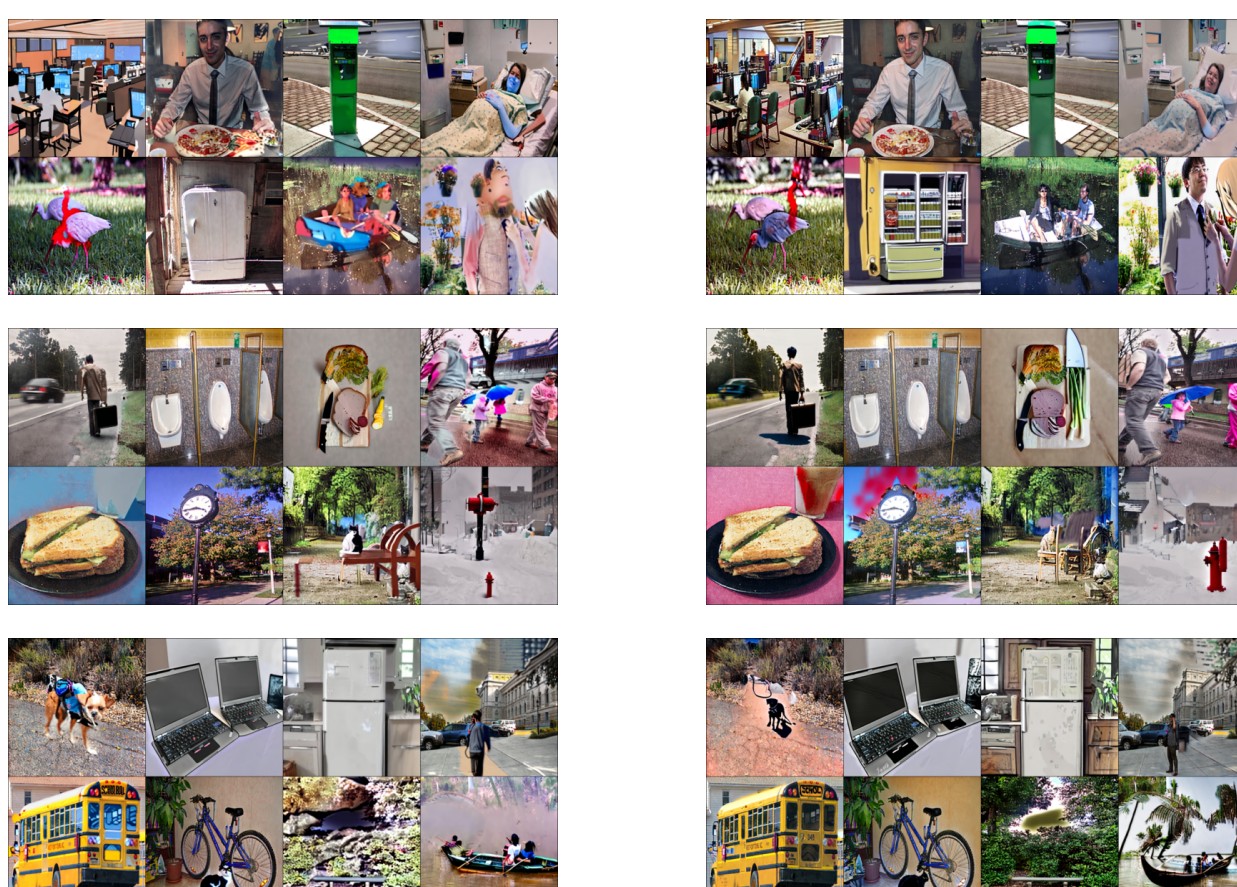

(a) With raw captions.  (b) With MLLM-generated descriptions.

Figure 4: Qualitative results of **SDXL-InstructPix2Pix**-generated images with Qwen2.5-VL-7B-generated descriptions and with raw captions from CIRCO validation split. We use 3.0 for image guidance scale, 7.5 for guidance scale, and 30 denoising steps.

## E  Prompts

We show the prompts used for MLLMs when generating composed descriptions based on reference images and text modifications. When using the prompts, images are converted to base64 format and then inserted into the prompts. We point out that MLLMs can also be employed to generate multiple descriptions from different aspects or views per query like in LDRE (Yang et al., 2024), which further enhances CIR performance at the cost of extra computational overhead.

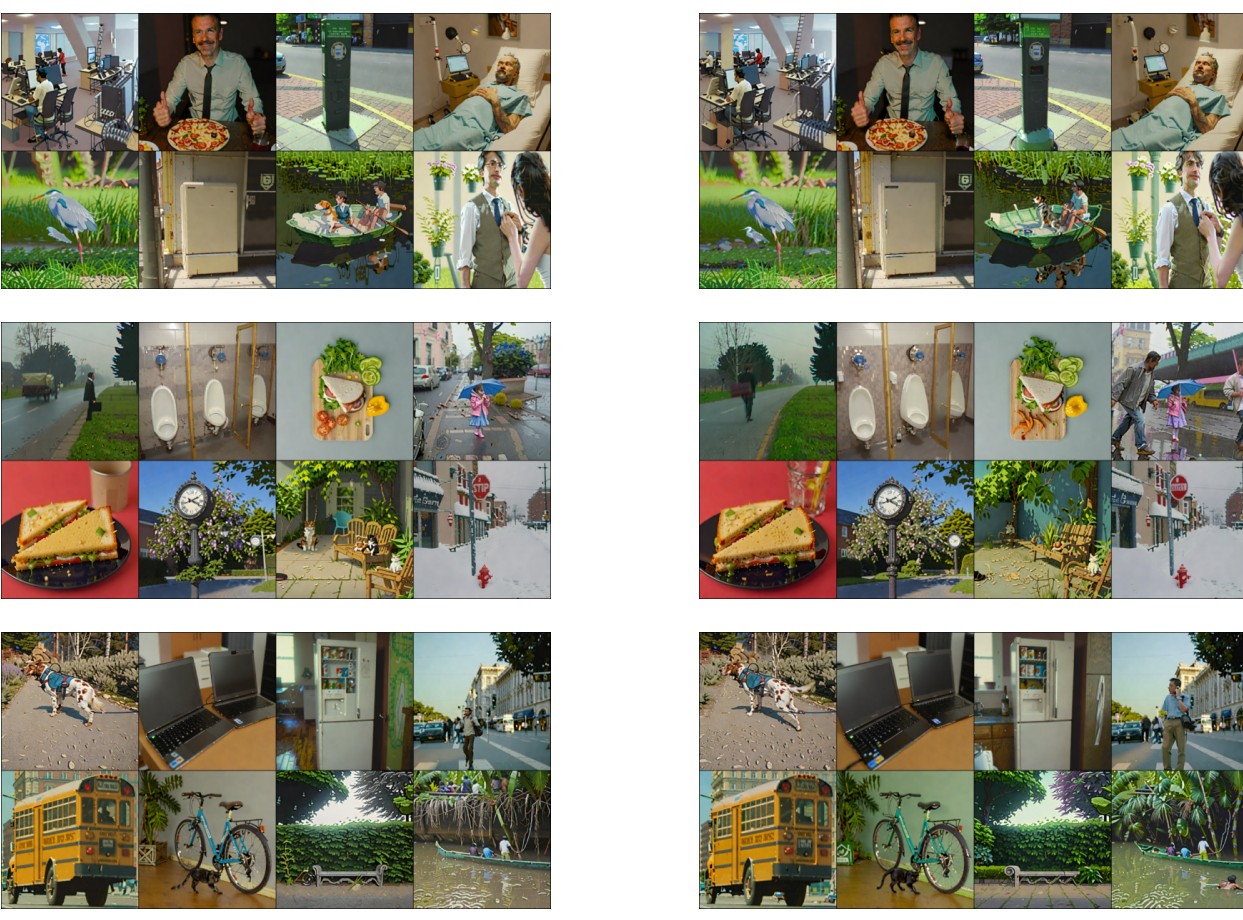

(a) With raw captions.                                    (b) With MLLM-generated descriptions.

Figure 5: Qualitative results of **SDXL-Turbo**-generated images with Qwen2.5-VL-7B-generated descriptions and with raw captions from CIRCO validation split. We use 0.5 for strength, 0.0 for guidance scale, and 30 denoising steps.

---

**Fashion-IQ**

You are an expert at visual perception and imagination of fashion items. Given a reference image of fashion items and modification instructions, mentally apply the changes and produce an accurate and complete natural-language description of the resulting fashion items. The modifications may describe direct attributes (e.g., "solid white with buttons"), comparisons (e.g., "longer sleeves," "lighter in color"), combined attributes (e.g., "black with a red cherry pattern and deep V neckline"), or negations (e.g., "no lace design"). Image: `base64_image`. Here are the modification instructions: `caption`. Focus on the fashion item and its attributes such as type, color, pattern, material, shape, fit, and style details. Ignore people and background from the image. Avoid imaginary things. Be specific and objective so that I can find targeting images based on your description solely without knowing the reference image or modification instructions. Do not use vague comparative terms like 'same/different/smaller/larger/shorter/longer/unchanged', etc. Instead, you should specify these differences clearly, like: another color instead of red (if no specific targeting color is mentioned), and a clear sky (if mentioned) instead of unchanged sky, etc. Now, describe how the final fashion item looks after applying the modifications. Write in 1 to 3 coherent sentences.

---

**CIRR**

You are an expert at visual imagination of real-world scenes. Given a reference image and modification instructions, mentally apply the modifications to the reference image and describe the resulting image in clear, complete English. Apply the modifications exactly as described, and ensure the final description reflects the scene after the changes. The modifications may include: 1. Cardinality: adjusting the number of objects (e.g., "only one bird remains"). 2. Addition: adding new objects or attributes (e.g., "add a red chair in the corner"). 3. Negation: removing elements (e.g., "remove the table"). 4. Direct Addressing: ensuring specific mentioned objects are clearly included. 5. Compare & Change: replacing one attribute with another (e.g., "same sofa but in leather"). 6. Comparative Statement: relative size, quantity, or intensity changes (e.g., "a larger group of people"). 7. Conjunction Statements: multiple modifications combined (e.g., "remove the tree and add two benches"). 8. Spatial Relations & Background: modifying positions, layout, or setting (e.g., "change the background to a beach"). 9. Viewpoint: adjusting perspective or framing (e.g., "zoom out to show the whole scene"). Image: `base64_image`. Here are the modification instructions: `caption`. Focus on the elements (like objects, people, and animals), their attributes (like color, size, shape, and quantity), spatial relations, and background. Avoid imaginary details and unnecessary repetitions. Be specific and objective so that I can find the targeting image from an image gallery based on your description solely without knowing the reference image or modification instructions. Do not use vague comparative terms like 'same/different/smaller/larger/shorter/longer/unchanged', etc. Instead, you should specify these differences clearly, like: another color instead of red (if no specific targeting color is mentioned), and a clear sky (if mentioned) instead of unchanged sky, etc. Write in 1 to 3 coherent sentences in English. Now, describe how the final image looks after applying the modifications.

---

## F    Failure Cases

By way of example, we show the failure cases when using OpenCLIP to retrieve top-5 images from CIRCO 123K-candidate image pool in Figure 6 when performing PᴇFᴜsᴇ (T→I), and Figure 7 when performing PᴇFᴜsᴇ (I→I). In Figure 6, Qwen-generated composed descriptions are used to retrieve target images. We observe that the generated queries, including queries 1, 2 and 3, cover the semantics of desired images, but might be more detailed than expected, causing mismatches with ground-truth images. For query 4, the generated textual query mistakenly treats the wrong objects as the object related to the modification, leading to wrong retrieval results. In Figure 7, all the generated images via `SDXL-InstructPix2Pix` do not

**CIRCO**

You are an expert at visual imagination of real-world scenes. Given a reference image and modification instructions, mentally apply the modifications and produce an accurate, detailed description of the resulting scene. Apply the modifications exactly as described, and describe the final scene after the changes. The modifications may involve: 1. Cardinality: adjusting the number of objects (e.g., "has two boxes"). 2. Addition: introducing new objects or attributes (e.g., "a child under the umbrella"). 3. Negation: removing elements (e.g., "shows no bike"). 4. Direct Addressing: ensuring a specific object is present (e.g., "next to a window"). 5. Compare & Change: altering attributes (e.g., "different color," "surrounded by flowers"). 6. Comparative Statements: relative size, number, or intensity (e.g., "more stickers," "larger crowd"). 7. Conjunction Statements: multiple edits at once (e.g., "surrounded by snow and trees are more bare"). 8. Spatial Relations & Background: positioning or environment changes (e.g., "skyscrapers in the background"). 9. Viewpoint: changes in perspective or framing (e.g., "shot from above"). Image: `base64_image`. Here are the modification instructions: `caption`. Focus on the objects, people, animals, attributes (color, size, shape, quantity), spatial relations, and background context. Be specific and objective. Avoid imaginary details not supported by the reference image or the modification. Do not use vague comparative terms like 'same/different/smaller/larger/shorter/longer/unchanged', etc. Instead, you should specify these differences clearly, like: another color instead of red (if no specific targeting color is mentioned), and a clear sky (if mentioned) instead of unchanged sky, etc. Write 1 to 3 complete and coherent sentences so that I can find targeting images based on your description solely without knowing the reference image or modification instructions. Now, describe how the final image looks after applying these modifications.

**GeneCIS**

You are an expert at visual imagination of real-world scenes. Given a reference image and modification instructions, you should distinguish the objects and their attributes from the reference image, abd then mentally apply the modifications to the reference image, and describe the objects and their attributes in the final image after the changes. Image: `base64_image`. Here are the modification instructions: `caption`. Avoid imaginary details not supported by the reference image or the modification. Write 1 to 3 complete and coherent sentences so that I can find targeting images based on your description solely without knowing the reference image or modification instructions. Now, describe the objects and their attributes after applying the modification.

fully match the modifications desired for target images, and most of the generated images are very similar to the original reference images, thereby degrading the retrieval performance.

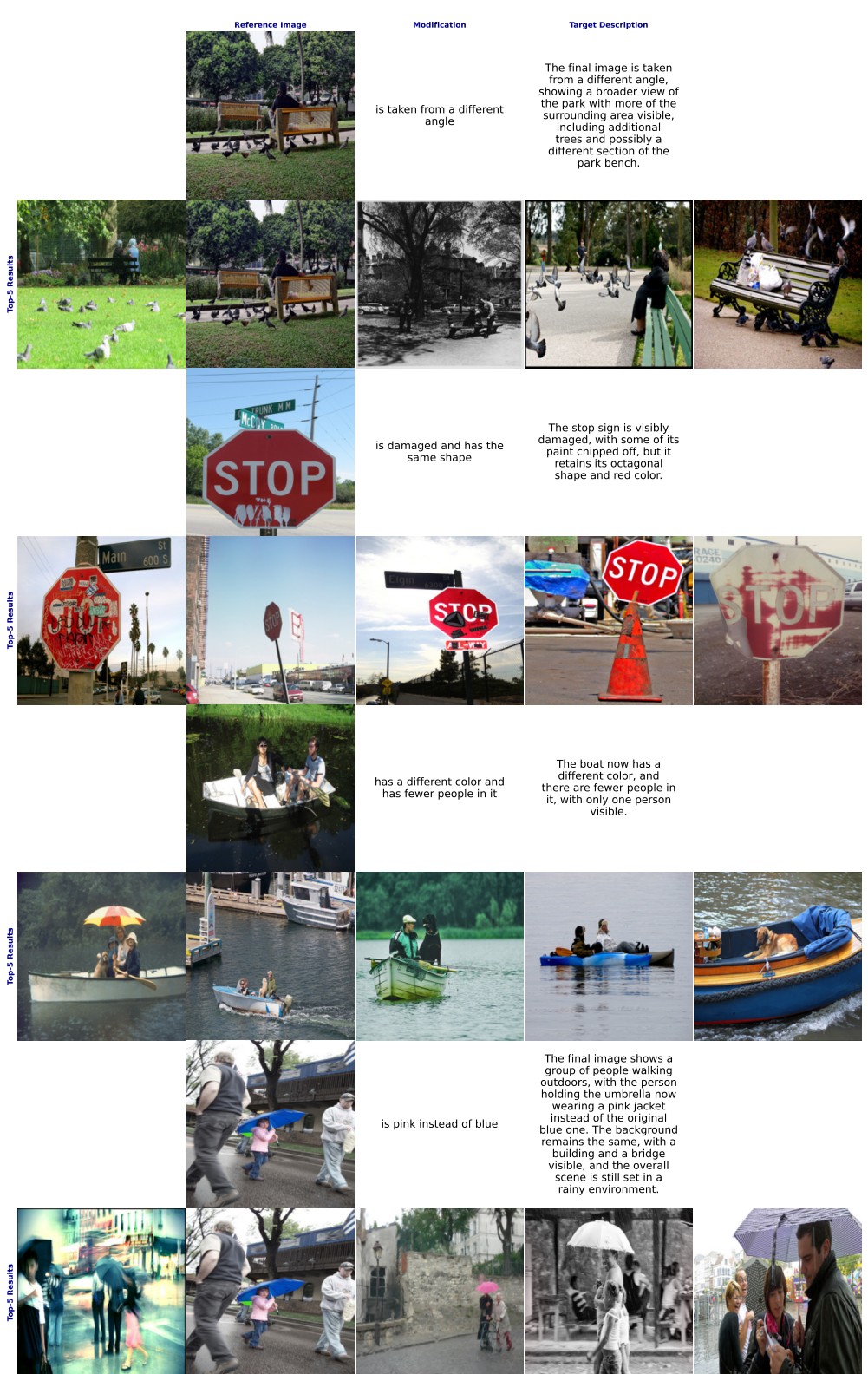

Figure 6: Failure cases of PeFuse (T→I). None of the top-5 results matches the corresponding query.

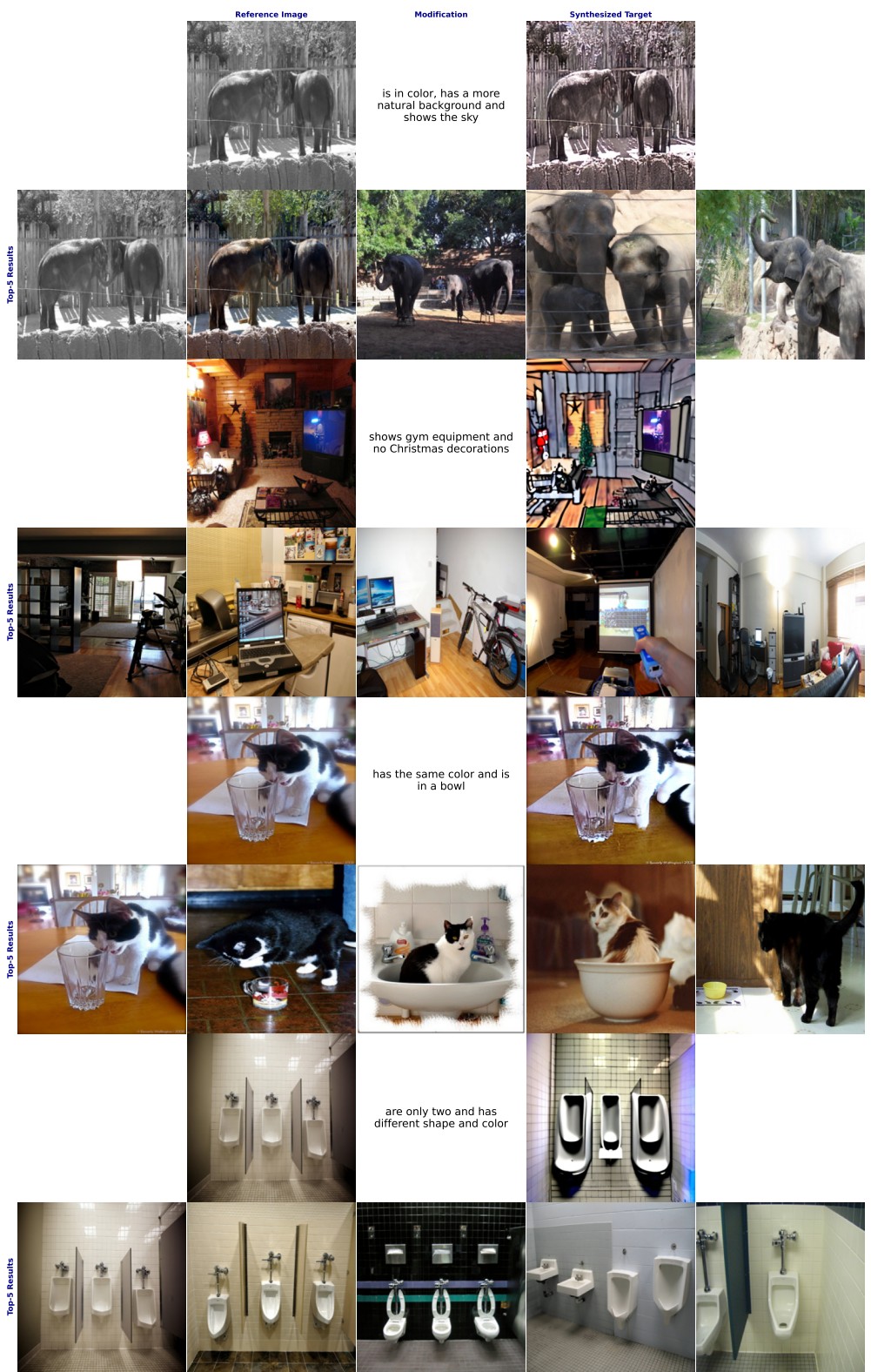

Figure 7: Failure cases of PEFUSE (I→I). None of the top-5 results matches the corresponding query.

