# OpenReview forum: "Training-Free Pseudo-Fusion Strategies for Composed Image Retrieval via Diffusion and Multimodal Large Language Models"
_TMLR — Under review for TMLR_

### Review · Reviewer_KQMc · 2026-03-26

**Summary Of Contributions:**

Summary: The paper studies zero-shot composed image retrieval (ZS-CIR) without task-specific training. It proposes PeFuse, a training-free “pseudo-fusion” framework that converts a composed query into four single-query retrieval settings: text-to-image, image-to-image, text-to-text, and image-to-text, using an MLLM or a diffusion model. The main empirical finding is that reformulating CIR as text-to-image retrieval is generally the strongest conversion, while image-to-image is consistently weaker; larger OpenCLIP backbones further improve performance.

Strength:
- The paper presents a clear and well-motivated training-free reformulation of composed image retrieval, converting multimodal queries into single-modality retrieval tasks using off-the-shelf generative and retrieval models.
- The empirical evaluation is generally comprehensive, covering four benchmarks, multiple retrieval backbones, and all four pseudo-fusion conversion strategies under a unified experimental framework.

Weakness:
- The methodology part is not clear enough. There is no problem formulation of the target task with definition of input and output. The illustration of two types of conversion is mixed in Figure 1, making the procedure of the method hard to follow. The retrieval step is also not clearly introduced.
- The empirical comparisons are not always strictly controlled, since several baseline results use different retrieval backbones, training settings, or evaluation subsets, which weakens the strength of cross-method performance claims. It is also confusing what is the difference between the evaluation setting of baseline method and that of ReFuse (including T→I, I→I, T→T, I→T), and is this type of comparison reasonable?

Questions:
- In figure 1, there are colored lines with the meaning of “T2I, T2T, I2I or I2T”, how about these black lines?

**Audience:**

Yes

**Audience Explanation:**

The audiences that work on Composed Image Retrieval or Content-based Image Retrieval will be interested in the findings of this paper.

**Broader Impact Concerns:**

There are not concerns that that would require adding a Broader Impact Statement.

**Claims And Evidence:**

Yes

**Claims Explanation:**

Generally, the claims and contributions in the paper is supported by accurate, convincing and clear evidence.

**Requested Changes:**

A problem formulation subsection to clearly introduce the target task.

An algorithm to introduce the procedure of the PeFuse.

Mark the notation in methodology part in Figure 1 for better understanding of the PeFuse method.

In some tables, there are “-” in some of the grids. Why these statistics are missed should be explained.

Small grammar mistakes in the paper, e.g. “we uses”, “can generated”.

---

> ### Author Response · Authors · 2026-06-22
> **Methodology and experiment clarification, and fixing of minor mistakes.**
>
> We thank the anonymous reviewer for the valuable comments. We have revised our manuscript to address the concerns raised, highlighted by an orange font color.
>
> 1. >  The methodology part is not clear enough. There is no problem formulating the target task with definition of input and output. The illustration of two types of conversion is mixed in Figure 1, making the procedure of the method hard to follow. The retrieval step is also not clearly introduced.
>
> Thanks for the suggestions. To clarify the audiences of our methods, we have improved the explanation in Section 3 and have added an algorithm figure that explicitly illustrates the process of each conversion mode, including both the uni-directional and the bi-directional ones. We also have explained in Section 4.1 how each dataset is evaluated.
>
> 2. > The empirical comparisons are not always strictly controlled, since several baseline results use different retrieval backbones, training settings, or evaluation subsets, which weakens the strength of cross-method performance claims. It is also confusing what is the difference between the evaluation setting of the baseline method and that of ReFuse (including T→I, I→I, T→T, I→T), and is this type of comparison reasonable?
>
> Thanks for the comment. To ensure fairness when comparing with CLIP-based baselines, we used the same retrieval models (CLIP/OpenCLIP) with the same vision backbone ViT-B-32. We also included the SigLIP model with the same backbone to provide additional insights. As PeFuse is a training-free method based on existing pre-trained models, no training was involved. For evaluation, we used the same official splits from each dataset as the baseline methods, and reported performance using the same metrics as previous work, so that our experiments are comparable with the baselines. Therefore we confirm that the comparison is reasonable across different conversion modes.
>
> 3. > In figure 1, there are colored lines with the meaning of “T2I, T2T, I2I or I2T”, how about these black lines? Mark the notation in methodology part in Figure 1 for better understanding of the PeFuse method.
>
> Thanks for the suggestion. We have clarified in the caption of Figure 1 that black arrows indicate the modality conversion flow.
>
> 4. > An algorithm to introduce the procedure of the PeFuse.
>
> Thanks for the suggestion. We have added an algorithm figure in Section 3 to summarize our method, so that readers can get a clearer picture of how **PeFuse** works.
>
> 5. > In some tables, there are “-” in some of the grids. Why these statistics are missed should be explained. Small grammar mistakes in the paper, e.g. “we uses”, “can generated”.
>
> Thanks for the suggestion. We have mentioned in the caption of Tables 2, 9, and 10 that missing results are not available from the original papers. We have also fixed the raised grammar mistakes as well.

---

### Review · Reviewer_bdFo · 2026-06-11

**Summary Of Contributions:**

This paper propose a training-free pseudo-fusion framework for zero-shot composed image retrieval. Different than the traditional method, it used pre-trained multimodal large language models and diffusion models to convert a composed query, consisting of a reference image and a text modification, into a single-modality retrieval task. It mainly studies four strategies: text-to-image, image-to-image, text-to-text, and image-to-text retrieval.
The main contribution is that it systematically compare the performance of these conversion strategies on several CIR benchmarks. The empirical result shows converting CIR to text-to-image is generally the most effective strategy. The paper also provides hyper-parameter and compute analysis for the used MLLM and diffusion models.
A key advantage of this framework is its simplicity, modularity, and lack of task-specific training. The experimental scope is also quite broad, providing valuable insights into which conversion path performs best. However, the method's innovation is limited, as it primarily combines existing pre-trained models without introducing new architectures, learning objectives, or theoretical analysis. Results largely depend on the choice of retrieval model and prompts.

**Audience:**

Yes

**Audience Explanation:**

I believe at least some individuals in TMLR's audience would be interested in the findings of this paper. The paper studies an important setting, zero-shot composed image retrieval, and provides a systematic empirical comparison of several training-free modality conversion strategies. Although the method itself mainly combines existing pre-trained models, the paper provides practical insights about which conversion strategy works better, how retrieval backbones affect performance, and how MLLM and diffusion hyper-parameters influence results. These findings could be useful for researchers interested in training-free retrieval pipelines.

**Broader Impact Concerns:**

Beyond the potential impacts common to image retrieval and generative models, I believe this framework will not have any other broader implications. Because it relies on pre-trained MLLM and diffusion models, it may inherit the biases or failure modes of these models. In practical retrieval systems, this could lead to biased or incorrect results, especially for sensitive visual attributes or underrepresented populations.

**Claims And Evidence:**

Yes

**Claims Explanation:**

The main claims are mostly supported by the experimental results. The paper provides results on several standard CIR benchmarks and compares different conversion strategies, which gives reasonable evidence that text-to-image reformulation is the strongest option among the tested settings.
However, the performance is strongly affected by the choice of retrieval model and prompts, so more controlled experiments would make the evidence more convincing.

**Requested Changes:**

I do not request major methodological changes. This is not critical to my recommendation, but it would strengthen the paper if the authors added a more detailed conceptual or theoretical discussion explaining why text-to-image reformulation tends to outperform the other conversion strategies.

---

> ### Author Response · Authors · 2026-06-22
> **Extra discussion of the causes of superiority of text-to-image retrieval**
>
> We thank the comments from the reviewer. To provide more insights of the superiority of text-to-image conversion mode via **PeFuse**, we have added more discussion of the potential causes in Section 4.3 for the phenomenon in our revised manuscript, highlighted by an orange font.

---

### Review · Reviewer_6Yjn · 2026-06-17

**Summary Of Contributions:**

This work studies the problem of **Composed Image Retrieval (CIR)**: given a reference image and a textual modification instruction, the goal is to retrieve a target image that satisfies the intended modification. The paper proposes **PEFUSE**, a pseudo-fusion framework that does not require downstream task-specific training. Its core idea is not to explicitly learn an image-text fusion module, but rather to use pretrained diffusion models and multimodal large language models for generative modality conversion, transforming the original composed query into several single-query retrieval problems.

I see the main strengths of this work as follows. First, CIR is a practical and challenging problem in image-text retrieval, with clear real-world application value. Second, the proposed method is intuitive and easy to understand; from my perspective, it is also novel.

The paper also has several weaknesses. First, the current claims of being “training-free” and “efficient” seem somewhat overstated. Although the method does not train a downstream CIR module, it heavily relies on large-scale pretrained MLLMs, diffusion models, and vision-language retrievers. Moreover, online inference may involve expensive generative model calls. Second, the experiments are not yet sufficient to show that the performance gains mainly come from the pseudo-fusion mechanism itself, rather than from a stronger backbone, prompt engineering, MLLM scale, or diffusion-model priors. Third, the paper lacks a systematic analysis of errors introduced by generative conversion. Both MLLMs and diffusion models may introduce hallucination, identity drift, over-editing, or under-editing, which are particularly important issues for CIR.

**Audience:**

Yes

**Audience Explanation:**

I believe that at least some TMLR readers would be interested in the findings of this paper, for three reasons.

First, CIR is a cross-modal retrieval task with clear practical value.

Second, the paper attempts to understand multimodal fusion from a different perspective. Traditional CIR methods usually learn an explicit image-text fusion representation, whereas this work reinterprets fusion as generative modality conversion followed by standard retrieval. Although this perspective is not entirely unprecedented, it is still informative. In particular, the paper’s systematic comparison of different conversion directions can help readers better understand the trade-offs among text rewriting, image generation, and cross-modal retrieval.

Third, the paper studies how foundation models can be used in retrieval pipelines in a post-training or non-training manner. This aligns well with the current machine learning community’s interest in MLLMs, diffusion models, training-free adaptation, and retrieval-augmented systems.

**Broader Impact Concerns:**

I do not believe there is a broader impact concern in this paper serious enough to prevent publication

**Claims And Evidence:**

No

**Claims Explanation:**

I believe that some of the paper’s core claims are not yet supported by sufficiently strong, clear, and convincing evidence, especially the claims regarding the effectiveness of pseudo-fusion, the training-free nature of the method, time efficiency, and superior or competitive performance.

First, the paper presents experimental results for PEFUSE on standard CIR benchmarks and compares different conversion strategies. These experiments suggest that the proposed framework is effective to some extent, particularly that the text-to-image retrieval formulation performs well under the authors’ setting. However, these results are still insufficient to show that pseudo-fusion itself is the main source of the performance gains. Since the method relies on several strong pretrained models, including MLLMs, diffusion models, and vision-language retrieval backbones, the observed improvements may instead come from stronger foundation models, prompt design, or related factors.

Second, the paper describes the method as training-free, but this statement should be made more precise. It is true that the method does not require downstream training on CIR triplets. However, the large-scale pretrained models it relies on already contain substantial visual, linguistic, and cross-modal knowledge. Therefore, a more accurate description would be “downstream-training-free” or “no task-specific training.”

Third, the evidence for time efficiency remains insufficient. If the method requires calling an MLLM or a diffusion model during online query processing, inference latency, memory usage, API cost, and batch throughput may all become practical deployment bottlenecks.

**Requested Changes:**

Critical changes
- The current use of the term “training-free” may be misleading. While the method does not train a downstream CIR module, it relies on large-scale pretrained MLLMs, diffusion models, and vision-language retrievers. The authors should clearly qualify this claim as “downstream-training-free” or “no task-specific training.”
- It is currently difficult to determine whether the performance gains of PEFUSE come from the proposed method itself, or from stronger foundation models or retrieval backbones. The authors should compare PEFUSE with strong baselines under the same retrieval backbone, the same gallery embeddings, and the same captioner/MLLM settings.
- If the paper claims time efficiency, the authors should report more fine-grained computational costs.
- The authors should include key ablation studies to isolate the actual contribution of the pseudo-fusion mechanism.
- The core challenge of CIR is to preserve the relevant information in the reference image while correctly applying the modifier. The authors should analyze the failure modes of MLLM/diffusion-based conversion.

Suggested changes
- MLLM decoding and diffusion sampling are both stochastic. I suggest reporting the mean, standard deviation, and worst-case performance across multiple random seeds or multiple generated candidates.

---

> ### Author Response · Authors · 2026-06-28
> **Term usage, fairness, and computational costs**
>
> We thanks the reviewer for the comments. We have addressed the concerns raised by the reviewer, and revised our manuscript accordingly. We listed the answers and corresponding questions below.
>
> 1. > The current use of the term “training-free” may be misleading. While the method does not train a downstream CIR module, it relies on large-scale pretrained MLLMs, diffusion models, and vision-language retrievers. The authors should clearly qualify this claim as “downstream-training-free” or “no task-specific training.”
>
> Thanks for the suggestion. Following the research literature, and in particular the baseline methods mentioned in our paper, we confirm that the use of “training-free” is reasonable in our case, because PeFuse relies on pre-trained models and does not involve any training or learning steps. We have added this clarification in the Abstract and Introduction sections.
>
> 2. > It is currently difficult to determine whether the performance gains of PEFUSE come from the proposed method itself, or from stronger foundation models or retrieval backbones. The authors should compare PEFUSE with strong baselines under the same retrieval backbone, the same gallery embeddings, and the same captioner/MLLM settings.
>
> Thanks for this comment. We agree that it is important to disentangle the effect of the proposed method from the effect of different foundation models or retrieval backbones. This is precisely what we intended to showcase in our experiments.
> We intentionally use retrieval models that process images at the same input resolution (224×224 px) and that are based on comparable ViT-Base architectures, consistent with the baseline methods considered in our paper, including LDRE, ImageScope, and CIReVL. Specifically, CLIP and OpenCLIP use ViT-B/32, while SigLIP2 uses ViT-B/16 because the available SigLIP2 ViT-B/32 checkpoint is designed for 256×256 input resolution rather than 224×224 px. Further, all selected backbones have comparable model sizes, which helps to control for model capacity. Following the standard evaluation protocol in CIR tasks, all methods are evaluated on the same dataset splits, against the same images, and using the same retrieval performance metrics. In addition to the CLIP/OpenCLIP settings used by prior baselines, we include SigLIP2 as a complementary retrieval backbone to better understand how the choice of retrieval model affects downstream CIR performance.
> Our results show that the retrieval backbone can indeed influence CIR performance. This further highlights the importance of selecting an appropriate retrieval model when converting multimodal composed queries into single-modality retrieval queries. We have clarified this point in Section 4.1 and explicitly stated the backbone, gallery, and evaluation settings used for comparison.
>
> 3. > If the paper claims time efficiency, the authors should report more fine-grained computational costs.
>
> Thanks for the comment. As PeFuse relies on pre-trained models, we acknowledged the extra overheads and indeed reported a latency analysis in Section 5 of the original submission. Specifically, we included in Table 6 both time and performance analyses of PeFuse (T→I) and PeFuse (I→I), as these two conversion modes involve the use of MLLMs and diffusion models. We also mentioned in Section 4.5.2 that tuning inference steps can have better time efficiency (see Figure 3 middle). In the revised paper, we have removed  the claim of efficiency that may cause confusion and have added a short discussion in Section 7 about the time latency incurred by the reliance on generative models.
> We also have updated Section 5 and Table 6 to explain more about fine-grained computational costs, including caption generation, image generation, and retrieval time. Specifically, for the baseline evaluations, the pipeline time directly reflects retrieval time, since no generation step is involved. For T→I conversion with MLLMs, caption-generation latency is captured by the inference time. For I→I conversion with diffusion models, the inference time represents the per-query image-generation time for both SDXL-InstructPix2Pix and SDXL-Turbo. Finally, when diffusion models are chained with Qwen2.5-VL-7B, the reported inference time includes both stages: first, generating the target-image description with the MLLM, and second, using that description as the refined prompt together with the reference image to generate an image with a diffusion model.

---

> ### Author Response · Authors · 2026-06-28
> **Ablation, failure cases, and stochasticity**
>
> 4. > The authors should include key ablation studies to isolate the actual contribution of the pseudo-fusion mechanism.
>
> Thanks for the suggestion. To better isolate the contribution of the proposed pseudo-fusion mechanism, we have added more ablations in Section 5. Specifically, we first establish baselines that do not rely on any generative models: (1) retrieving target images using only the reference image representations, (2) retrieving target images using only the modification representations, and (3) retrieving target images using the normalized combination of the reference image and modification representations. These baselines allow us to quantify the performance achievable without PeFuse. We then compare these results with PeFuse using different generative backbones, including three MLLMs for the PeFuse (T→I) setting and two diffusion models for the PeFuse (I→I) setting on the CIRCO validation split in Table 6. The results show that using PeFuse consistently improves retrieval performance over all non-generative baselines. Furthermore, the performance differences across MLLMs and diffusion models demonstrate that the choice of generative backbone influences the quality of the pseudo-fused representations, highlighting the importance of selecting appropriate generative models. We also report the corresponding inference latency to illustrate the trade-off between retrieval accuracy and computational cost.
> These additional ablation studies clearly disentangle the contribution of each component of  PeFuse from that of not converting multimodal queries into single-modality queries. We have updated the experimental results and corresponding discussion in Section 5 for clarification. We also attached the table as follows:
> | Method | Generative Models | Memory (MB) | Inference Time (s) | Pipeline Time (s) | Avg mAP (%) |
> |---|---:|---:|---:|---:|---:|
> | Modification→Target | - | - | - | 0.48 (± 0.02) | 7.36 |
> | Reference→Target | - | - | - | 0.48 (± 0.02) | 3.48 |
> | Reference+modification→Target | - | - | - | 0.49 (± 0.02) | 6.61 |
> | PeFuse (T→I) | Qwen2.5-VL-3B | 7162 | 0.69 (± 0.03) | 1.70 (± 0.11) | 25.15 (± 0.80) |
> | PeFuse (T→I) | Qwen2.5-VL-7B | 15818 | 0.41 (± 0.03) | 1.70 (± 0.14) | 24.97 (± 0.37) |
> | PeFuse (T→I) | LLaVA-1.5-7B-hf | 13472 | 0.70 (± 0.04) | 3.01 (± 0.10) | 17.11 (± 1.09) |
> | PeFuse (I→I) | SDXL-Instr. | 6725 | 3.95 (± 0.02) | 6.24 (± 0.05) | 5.22 (± 0.08) |
> | PeFuse (I→I) | SDXL-Turbo | 6725 | 1.25 (± 0.02) | 3.07 (± 0.27) | 7.40 (± 0.56) |
> | PeFuse (I→I) | Qwen2.5-VL-7B + SDXL-Instr. | 15818 + 6725 | 4.40 (± 0.03) | 6.33 (± 0.23) | 5.33 (± 0.70) |
> | PeFuse (I→I) | Qwen2.5-VL-7B + SDXL-Turbo | 15818 + 6725 | 1.69 (± 0.05) | 3.49 (± 0.33) | 7.18 (± 0.41) |
>
> 5. > The core challenge of CIR is to preserve the relevant information in the reference image while correctly applying the modifier. The authors should analyze the failure modes of MLLM/diffusion-based conversion.
>
> Thanks for the suggestion. In the original submission, we already included MLLM-based failure cases in Appendix F. We have further added diffusion-based failure cases and expanded the discussion of their conversion errors. These updated analyses better highlight failure modes, such as losing key reference-image attributes, incorrectly applying the modifier, and introducing irrelevant details, thereby providing deeper insight into the limitations of conversion-based CIR methods.
>
> 6. > MLLM decoding and diffusion sampling are both stochastic. I suggest reporting the mean, standard deviation, and worst-case performance across multiple random seeds or multiple generated candidates.
>
> Thanks for this suggestion. In our experiments, we follow the standard protocols of prior work by fixing the random seeds for both MLLM-based decoding and diffusion-based sampling, while keeping all hyperparameters identical across runs to ensure reproducibility.
> In addition, we report multi-run statistics to quantify the effect of stochasticity. As shown in Table 6 and Figures 2 and 3, we provide the mean and standard deviation across different random seeds. The results indicate that retrieval performance is generally stable across seeds, suggesting that the improvements of our method are not caused by a single favorable generation. We have clarified this experimental setting and expanded the corresponding discussion in Sections 4.5 and 5 of the revised manuscript.

---

> > ### Comment · Reviewer_6Yjn · 2026-07-14
> >
> > Thank you for your response. My concerns have been fully addressed.

---

### Author Response · Authors · 2026-06-28

We would like to express our gratitude to the anonymous reviewers for their valuable comments. We believe that our paper has improved further as a result. We have addressed the questions raised and explained how we have implemented the changes in the revised version of the paper. We have used an orange font color to highlight the changes, to facilitate re-review.